# Tidal wetland resilience to sea level rise increases their carbon sequestration capacity in United States

Faming Wang [1,2,3,4,5,8], Xiaoliang Lu[6,8], Christian J. Sanders[4,7] & Jianwu Tang [4,5*]

Coastal wetlands are large reservoirs of soil carbon (C). However, the annual C accumulation rates contributing to the C storage in these systems have yet to be spatially estimated on a large scale. We synthesized C accumulation rate (CAR) in tidal wetlands of the conterminous United States (US), upscaled the CAR to national scale, and predicted trends based on climate change scenarios. Here, we show that the mean CAR is $161.8 \pm 6$ g C m$^{-2}$ yr$^{-1}$, and the conterminous US tidal wetlands sequestrate 4.2–5.0 Tg C yr$^{-1}$. Relative sea level rise (RSLR) largely regulates the CAR. The tidal wetland CAR is projected to increase in this century and continue their C sequestration capacity in all climate change scenarios, suggesting a strong resilience to sea level rise. These results serve as a baseline assessment of C accumulation in tidal wetlands of US, and indicate a significant C sink throughout this century.

[1] Xiaoliang Research Station for Tropical Coastal Ecosystems and Key Laboratory of Vegetation Restoration and Management of Degraded Ecosystems, South China Botanical Garden, Chinese Academy of Sciences, Guangzhou 510650, P.R. China. [2] Center of Plant Ecology, Core Botanical Gardens, Chinese Academy of Sciences, Guangzhou 510650, P.R. China. [3] Southern Marine Science and Engineering Guangdong Laboratory (Guangzhou), Guangzhou 511458, P.R. China. [4] State Key Laboratory of Estuarine and Coastal Research and Institute of Eco-Chongming, East China Normal University, Shanghai 201100, P. R. China. [5] The Ecosystems Center, Marine Biological Laboratory, Woods Hole, MA 02543, USA. [6] State Key Laboratory of Soil Erosion and Dryland Farming on the Loess Plateau, Institute of Soil and Water Conservation, Northwest A&F University, Yangling, Shaanxi, China 712100. [7] National Marine Science Centre, School of Environment, Science and Engineering, Southern Cross University, Coffs Harbour, NSW 2450, Australia. [8] These authors contributed equally: Farming Wang, Xiaoliang Lu. *email: jtang@mbl.edu

Tidal wetlands, including tidal marshes and mangroves, contain long-term soil organic carbon (C) of which continuously sequester atmospheric carbon dioxide at rates 10–100s times higher than terrestrial forests[1]. Despite only covering 2% of the earth's ocean surface, global tidal wetlands have been roughly estimated to bury ~ 116 Tg C yr$^{-1}$, which accounts for over 50% of annual C burial in the ocean[2,3]. The C stocks and fluxes in these intertidal environments are collectively regarded as coastal wetland blue carbon[1].

Soils in these blue C ecosystems, unlike terrestrial ecosystems, do not become saturated with C as they accrete vertically under the impact of rising sea levels[1]. Thus, the rate of sediment C sequestration can be maintained as long as the soil accretion continues to keep pace with sea level rise. In terrestrial ecosystems, although fast C restoration rates were recorded in vegetations and/or soils after disturbance[4,5], these rates were maintained only on a decadal scale[5], and would reach a relative saturation point after restorations when the net C sequestration rate approximates to zero owing to increased soil and vegetation respiration. Therefore, the longevity of tidal C sinks has a greater potential than terrestrial C sinks in mitigating climate change over longer periods. There is an increasing global interest in coastal wetlands as targets for greenhouse gas emission offset projects through the preservation and restoration of these ecosystems to increase future C sequestration[1,6].

Rates of C sequestration in tidal wetlands depend upon their vertical sediment accretion rate (SAR) and soil C density. In healthy tidal wetlands, SAR generally keeps pace with relative sea level rise (RSLR) by the accumulation of mineral and organic sediment[7–10]. In a New England salt marsh, Redfield and Rubin[11] reported well-maintained soil elevation equilibrium with sea level for the previous 4000 yrs. Morris et al.[7] developed a model to predict the response of coastal wetlands to rising sea level, and reported that limiting rate of RSLR on the southeast coast of US was predicted to be 12 mm yr$^{-1}$, which is 3.5 times greater than the current long-term rate of RSLR, indicating that SAR in the coastal wetlands of this region could keep pace with current RSLR. In comparison with SAR, the C density usually varied substantially across different vegetation types and regions as revealed in the 2013 compilation by the International Panel on Climate Change (IPCC)[12]. Although some studies have reported a negative relationship between soil C density and mean annual temperature[13], this relationship is very weak, and studies have yet to explain the variability in carbon density across different regions. Higher decomposition rates of soil organic matter usually accompany higher temperatures. However, higher temperatures also enhance plant productivity, which would partially compensate the loss of C through increased decomposition[14]. Therefore, a more thorough understanding of the drivers of tidal wetlands C sequestration is highly needed.

The USA has vast tracts of tidal wetlands, with a total area in the range from 22,000 to 26,000 km$^2$[15–18]. Hinson et al.[16] estimated that there was 1153–1359 Tg of SOC stored in the upper 0–100 cm of soils in 24945.9 km$^2$ tidal wetlands across the conterminous United States. This value was over 10% of total the C stocks in top 0–120 cms soils of all US wetlands estimated by Nahlik and Fennessy[17], whereas tidal wetlands only accounted for 6.5% of total area of all wetlands in the conterminous United States (384,000 km$^2$), highlighting the importance of tidal wetlands in storing carbon. Moreover, this value may still greatly underestimate the soil C stocks in tidal wetlands as their soil profiles are known to reach six to 13 m in depth, and their C density has generally been found to be consistent with depth[11,17,19], indicating their disproportional contribution to the total wetland C storage when the entire soil profile is considered.

Although the total C stocks of these tidal wetlands across the conterminous United States have been estimated by several independent studies[16–18,20], no study has provided a detailed spatial distribution of their C sequestration or C accumulation rates on a national scale. Hinson et al.[16] estimated the C sequestration rate to be ~ 1.5 Tg C yr$^{-1}$ in the conterminous United States tidal wetlands. This estimation was obtained by simply assuming a C burial rate of 60 g m$^{-2}$ yr$^{-1}$ and the C density to be 0.03 g cm$^{-3}$ for all tidal wetlands. However, these estimates may be highly underestimated[13,21]. Using 154 sites across the globe, Chmura et al.[13] found that the average soil C density is 0.055 g cm$^{-3}$ and 0.039 g cm$^{-3}$ in mangrove and salt marsh, respectively. According to the US Inventory of Greenhouse Gas Emissions and Sinks, US tidal wetlands sequestered 3.3 Tg C yr$^{-1}$ during the period from 2005 to 2016[20]. This value was derived from a synthesis of peer-reviewed literature, but the details in calculating this value were not provided. Therefore, a spatially explicit database that details tidal wetland C sequestration rates is needed for Tier 2 estimation of the conterminous United States. Furthermore, the balance of controlling factors and their relative contributions to C sequestration is not well known in the tidal wetlands. Identifying the controlling factors of C accumulation dynamics is critical for understanding the fate of the C buried by these wetlands under future climate change and human activities.

In this study, we aim to estimate the C accumulation rate in different regions in the conterminous United States from compiled data, evaluate the factors controlling the spatial patterns of sediment accretion and C accumulation, and predict the future C fluxes in these tidal wetlands. We show that tidal wetlands would continue their C sequestration capacity in all representative concentration pathways scenarios even under the most-restricted lateral accommodation space availability in the conterminous United States, which suggests a strong resilience to sea level rise.

## Results

**National tidal wetland accretion and C accumulation.** The average C accumulation rates (CAR) in the conterminous United States tidal wetland were estimated to be 161.8 ± 6 g C m$^{-2}$ yr$^{-1}$ (Table 1). The National Wetland Inventory data indicates that there is a total of 2.59 million hectares of tidal wetlands (Fig. 1), with over 97% distributed along the east coast and Gulf-Bay (Fig. 2 and Table 1). By extrapolating the national mean CAR to all tidal wetlands, we estimated that tidal wetlands in the conterminous United States accumulated 4.19 Tg yr$^{-1}$ C. By summing the regional level C accumulation data (Table 1), the tidal wetlands accumulated ~4.97 Tg yr$^{-1}$. Moreover, based on the spatial interpolation method, the C accumulation in the 325,255 tidal wetland polygons listed in the NWI database was tallied (Fig. 1), and the sum of all the conterminous United States tidal wetlands C accumulation was 4.59 Tg r$^{-1}$. Therefore, we estimated that the conterminous United States accumulated C in the range of 4.2–Tg yr$^{-1}$, depending on different upscaling methods.

**Geographic patterns.** Soil C density ranged from 0.032 g C cm$^{-3}$ to 0.045 g C cm$^{-3}$, with the highest rate at the Mid Atlantic region, and the lowest at the California region. We collected 343 unique measurements of soil accretion rate (SAR) across the conterminous United States. A comparison with the local rate of RSLR indicated that 130 of 343 sites were submerging, whereas the remaining sites have higher SAR than the local RSLR rate (Fig. 3a). The lower Mississippi region had the highest SAR (8.89 mm yr$^{-1}$, Fig. 3b & Table 1) among the seven regions, and was significantly higher than the rate in all the other regions except Texas-Gulf (5.3 ± 0.9 mm yr$^{-1}$, Table 1). The lower

**Table 1 The C accumulation rate (CAR), C density, sediment accretion rate (SAR), tidal wetland area, and annual CAR in different regions of the conterminous United States (mean ± s.e.m.).**

| Regions | CAR (g C m$^{-2}$ yr$^{-1}$) | C Density (g cm$^{-3}$) | SAR (mm yr$^{-1}$) | RSLR (mm yr$^{-1}$) | Area (km$^2$) | C acc. by regions (Tg yr$^{-1}$) | C acc. By NWI (Tg yr$^{-1}$) |
|---|---|---|---|---|---|---|---|
| Mid Atlantic | 176.5 ± 14[b] | 0.045 ± 0.002[a] | 4.46 ± 0.33 [b] | 3.82 | 3844 | 0.710 ± 0.055 | 0.62 |
| N | 85 | 94 | 90 | | | | |
| New England | 151.3 ± 11[bc] | 0.039 ± 0.001[ab] | 3.72 ± 0.22 [b] | 2.78 | 512 | 0.078 ± 0.006 | 0.07 |
| N | 64 | 70 | 78 | | | | |
| South Atlantic-Gulf | 123.6 ± 11[c] | 0.034 ± 0.002[b] | 3.95 ± 0.34[b] | 2.95 | 10359 | 1.176 ± 0.105 | 1.13 |
| N | 69 | 69 | 68 | | | | |
| Lower Mississippi | 271.9 ± 18[a] | 0.034 ± 0.002[b] | 8.89 ± 0.44[a] | 8.91 | 8193 | 2.562 ± 0.154 | 2.13 |
| N | 43 | 47 | 43 | | | | |
| Texas-Gulf | 237.8 ± 16[abc] | 0.039 ± 0.005[ab] | 5.30 ± 0.9[b] | 6.59 | 2340 | 0.372 ± 0.025 | 0.57 |
| N | 2 | 2 | 2 | | | | |
| California | 103.8 ± 8[c] | 0.032 ± 0.002[b] | 4.79 ± 0.43 [b] | 1.98 | 269 | 0.027 ± 0.002 | 0.027 |
| N | 36 | 40 | 40 | | | | |
| Pacific Northwest | 110.2 ± 6[c] | 0.037 ± 0.004[ab] | 3.27 ± 0.39 [b] | 0.98 | 373 | 0.041 ± 0.002 | 0.04 |
| N | 11 | 11 | 11 | | | | |
| Total CONUS | 161.8 ± 6 | 0.038 ± 0.001 | 4.72 ± 0.16 | – | 25892 | 4.966 | 4.59 |
| N | 310 | 333 | 343 | | | | |

N number of observations. Different lowercase letters indicated a significant difference among vegetations (Tukey HSD)

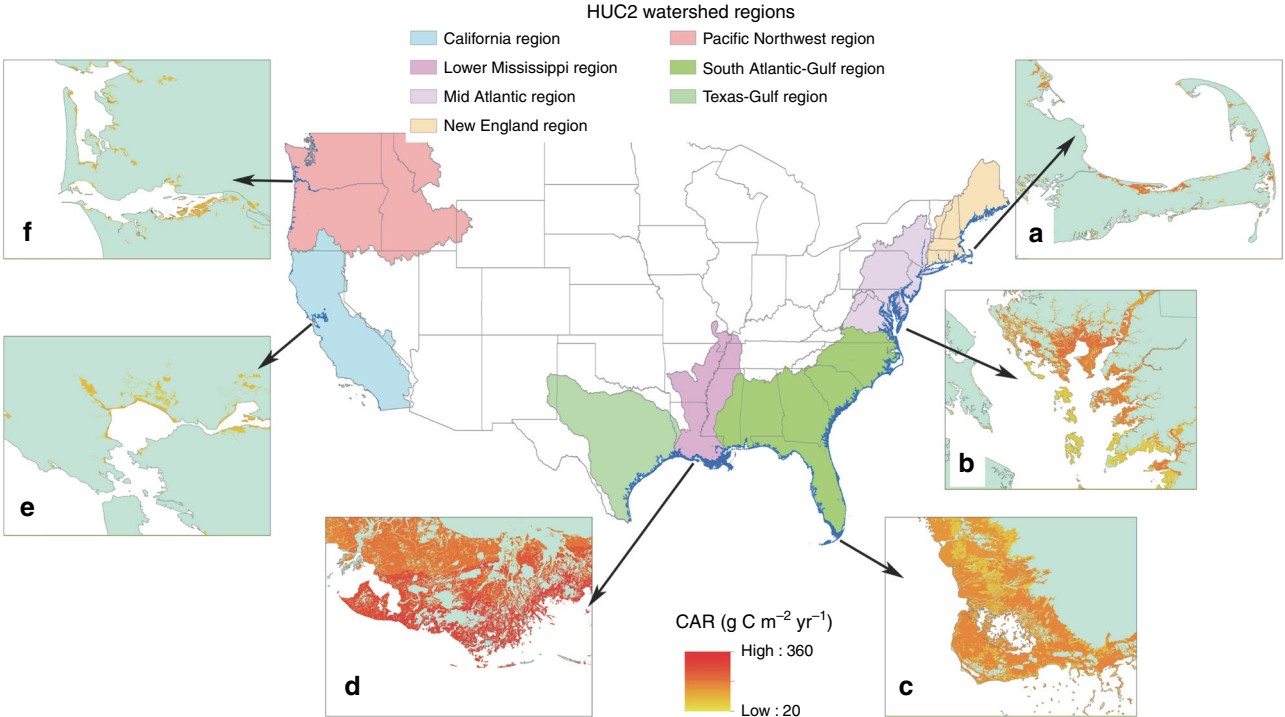

**Fig. 1** The tidal wetlands distribution (blue color) on the seven watershed regions along the conterminous United States and C accumulation rates (CAR) of tidal wetlands in six selected regions. **a** Cape Cod in Massachusetts, **b** the Chesapeake Bay in Maryland, **c** southern tip of Florida, **d** the Mississippi River Delta in Louisiana, **e** the Bay Area in California, and **f** the Columbia River Delta in Oregon.

Mississippi was also the region where SAR (8.89 mm yr$^{-1}$) was similar to the regional RSLR (8.91 mm yr$^{-1}$, Table 1). The CAR showed high variability among the regions (Table 1 and Fig. 2). The highest CAR rate was observed in the Lower Mississippi region (271.9 ± 16 g C m$^{-2}$ yr$^{-1}$), which was significantly higher than those in other regions except for the Texas-Gulf (Table 1), where there were only two reported data.

The lower Mississippi region had 32% of the tidal wetlands area in the conterminous United States (8193 km$^2$), whereas accounting for ~ 46 % of the annual C sequestration (2.13 Tg C yr$^{-1}$,

Table 1). The South Atlantic-Gulf was found to be the second largest C sequestration region (1.13 Tg C yr$^{-1}$). More than half of annual US tidal wetland C sequestration occurred in the Gulf-Bay (Lower Mississippi and Texas-Gulf: 2.70 Tg yr$^{-1}$), followed by East Coast (Including New England, Mid Atlantic and South Atlantic-Gulf) with 1.96 Tg yr$^{-1}$, with the lowest in the West Coast (0.07 Tg yr$^{-11}$).

Among the coastal states, LA had the highest CAR (274.2 ± 17 g C m$^{-2}$ yr$^{-1}$), which was significantly higher than the values in CA, CT, FL, GA, MA, ME, NC, NJ, VA, and WA (Fig. 4). TX only

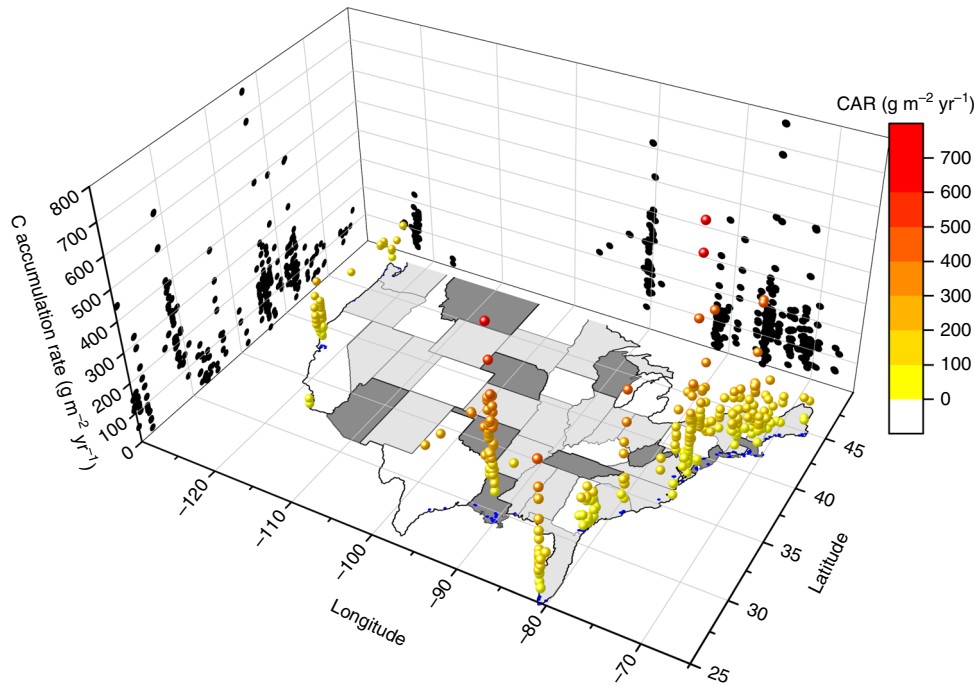

**Fig. 2** The 3D scatter plot of all soil C accumulation rate points across the tidal wetlands in the conterminous United States. The black dots on side walls are the projection of each data point to Latitude and Longitude, respectively.

contains two reported data (Fig. 2), however, the average of the data represent the second highest value ($237.8 \pm 16$ g C m$^{-2}$ yr$^{-1}$) among the coastal states. The lowest average CAR ($62 \pm 7.8$ g C m$^{-2}$ yr$^{-1}$) was found in GA. LA also had the largest tidal wetland area among these states, up to 9089 km$^2$. Combined with its high CAR (Fig. 3), LA accumulated 2.34 Tg C yr$^{-1}$ (Fig.5). FL had the second largest wetland area, most of which was mangrove, and its total annual C accumulation was estimated to be near 0.59 Tg yr$^{-1}$ (Fig. 5).

**Variations in CAR and regional factors**. The CAR was the highest in brackish, followed by tidal freshwater wetlands, and the lowest at mangroves ($151.5 \pm 16$ g C m$^{-2}$ yr$^{-1}$, Table 2). Owing to the high variations within each vegetation type, no statistical difference was found among the four types (Table 2). However, vegetation types significantly affected both SAR and bulk density ($p < 0.05$ for both, Table 2). Tidal freshwater wetlands had the highest SAR ($6.13 \pm 0.63$ mm yr$^{-1}$, Table 2), which is statistically higher than the rate in the salt marsh. Salt marsh had significantly lower C density than brackish and mangrove (Table 2). There was no statistical difference among vegetation types in terms of organic or mineral sedimentation rates. However, mangroves had the lowest mineral sedimentation rate ($586 \pm 117$ g m$^{-2}$ yr$^{-1}$) among all the vegetation types, and the value was only approximately half of the rates found in the other three vegetation types (Table 2). Owing to the distinguished elevation distribution, salt marsh can be divided into high marsh and low marsh. We also compared the difference between high marsh and low marsh and found no significant difference in CAR and organic sedimentation rate, but low marsh had significantly higher mineral sedimentation than high marsh ($p < 0.05$).

The CAR was positively correlated with SAR, C density, and RSLR, whereas it was negatively associated with salinity. Overall, salinity only explained <3% variability in CAR and C density. RSLR showed a strong correlation with CAR and SAR ($r = 0.40$ and 0.48, respectively, Table 3 and Fig. 3). Both Prcp and Tair had positive relationships with CAR ($p < 0.01$ for both, Table 3). SAR was positively associated with Tair ($r = 0.32$, $p < 0.01$), but had no

significant relationship with Prcp. The CAR decreased with increasing Latitude ($r = -0.18$, $p < 0.01$, Table 3, Fig. 2), whereas it had a much weaker correlation coefficient with Longitude ($r = 0.11$, $p < 0.05$, Table 3, Fig. 2).

Owing to the close correlation between RSLR and CAR, pathway analysis was further conducted to investigate the direct and indirect effects of RSLR on CAR (Fig. 6). The final model indicated that the effect of RSLR on CAR was mostly through the indirect pathways mediated by organic sedimentation and SAR. Furthermore, the pathway analysis also indicated that the RSLR had a more-significant positive effect on organic sedimentation than mineral sedimentation (coefficient 0.32 vs 0.10, Fig. 6)

**Prediction of tidal wetlands C sequestration**. We predict the future CAR based on the modeled future RSLR data[22] for each site. Together with the predicted tidal wetland area change[23], we calculate the future tidal wetlands C sequestration under different IPCC scenarios in the conterminous United States (Fig. 7). Under business-as-usual Representative Concentration Pathways (RCP) 8.5 scenario, the conterminous United States tidal wetlands would double its C sequestration amount to 9.40 Tg C yr$^{-1}$ (95% CI: 6.70–12.1 Tg C yr$^{-1}$) in 2100 if their lateral accommodation space is not restricted by human activities to population density higher than 300 peoples/km$^2$ (Fig. 7). Even at the most-restricted saturation (no lateral space available for the region with a population density higher than five people km$^{-2}$), the tidal wetlands still sequester 4.32, 4.29, and 4.13 Tg C yr$^{-1}$ under RCP 2.6, RCP 4.5, and RCP 8.5 scenarios, respectively (Fig. 7).

**Discussion**
Compared with previous estimations, the rates in this study show both differences and common features. In salt marsh, our average CAR ($154.3 \pm 8$ g C m$^{-2}$ yr$^{-1}$) was much lower than the global mean value (244 g C m$^{-2}$ yr$^{-1}$) reported by Ouyang and Lee[24], but similar to the rate (151 g C m$^{-2}$ yr$^{-1}$) reported by Duarte et al.[2]. This difference can be explained by the latitude range: both of this study and the one by Duarte et al.[2] has similar latitudinal

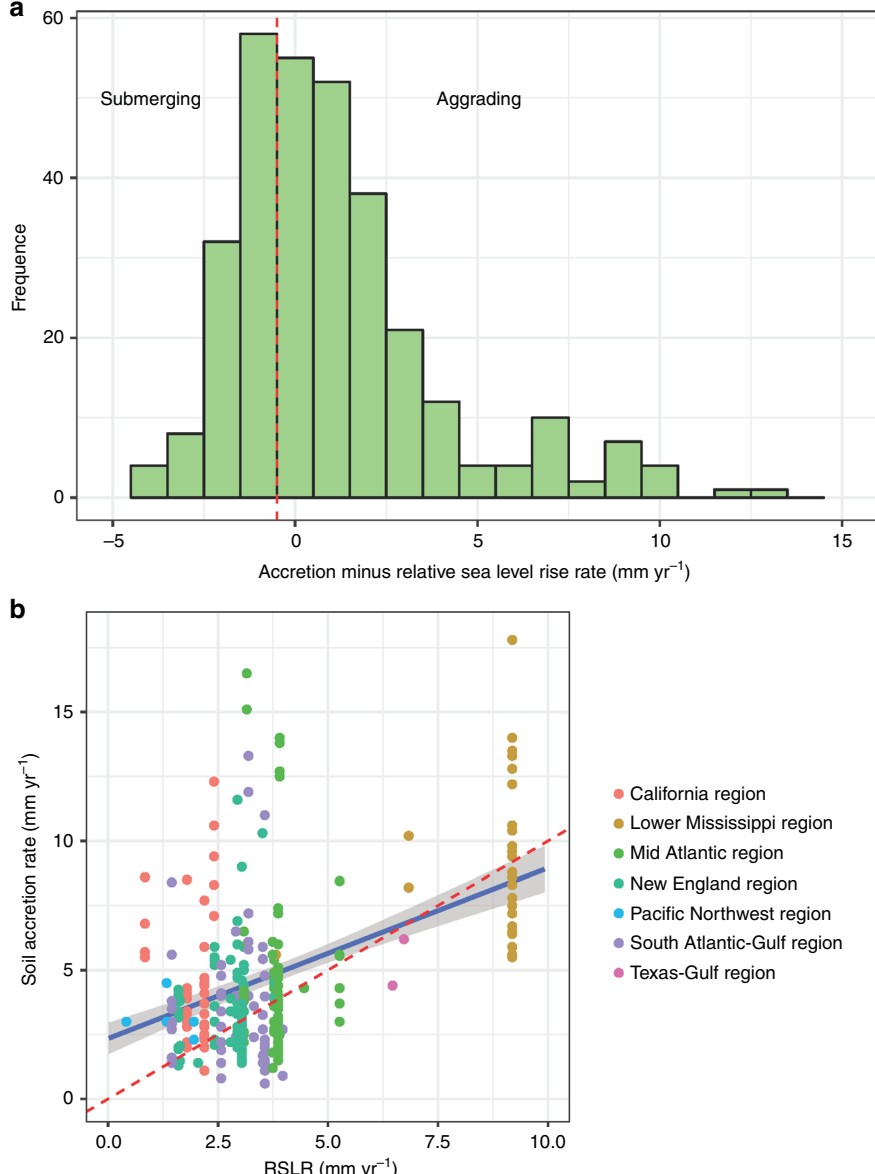

**Fig. 3** Analysis of soil accretion rate and relative sea level rise rate (RSLR) in tidal wetlands of the conterminous United States. **a** Frequency distribution of soil accretion rate relative to RSLR, where negative rates indicate submerging wetlands and positive rates represent aggrading wetlands. **b** Comparison between RSLR and soil accretion rate (SAR) in different regions of CONUS. The red dashed line represents an equilibrium condition where wetlands area building vertically at the same rate that RSLR. The blue line indicates the linear relationship between SAR and RSLR with the shadow as 95% confidence intervals.

ranges (25–50°N), whereas the global averages estimated by Ouyang and Lee[24] included data from other regions, like Asia-Pacific, Arctic, and Australasia. In mangroves, Breithaupt et al.[21] reported the geometric mean of the global CAR database to be 163.3 g C m$^{-2}$ yr$^{-1}$, which is similar to that in this study (151.5 ± 16 g C m$^{-2}$ yr$^{-1}$).

Lousiana (LA) had the most extensive tidal wetlands and the highest C accumulation amount among all the coastal states. DeLaune and White[25] estimated that 2.96 Tg C was sequestered annually in LA through vertical accretion. This value is ~30% higher than our estimate in LA (2.34 Tg C). DeLaune and White[25] obtained this value by multiplying a 300 g C m$^{-2}$ yr$^{-1}$ C burial rate with the coastal wetlands area in 1990, which was 9889 km$^2$. Although their C burial rate is similar to what we report here, the total wetland area is over 10% less in our study. In this study, we

used the most recent NWI data set (2017), which indicated that the tidal wetlands in LA had shrunk to 9086 km$^2$.

To our knowledge, there are no detailed reports on the national-wide tidal wetland C sequestration rate in the conterminous United States prior to the present study. However, some regional estimates of CAR were available for comparison. From the 104 measurements of CAR in salt marsh in the conterminous United States, Ouyang and Lee[24] estimated that there was 1.39 Tg C yr$^{-1}$, 0.36 Tg C yr$^{-1}$, and 2.52 Tg C yr$^{-1}$ C accumulated in salt marshes of East Coast, West Coast, and Gulf-Bay, respectively. They also reported that the highest CAR rates occurred in the Gulf-Bay regions, especially the LA salt marshes. This spatial pattern is in agreement with our results that the Lower Mississippi region has the highest average CAR (271 ± 18 g C m$^{-2}$ yr$^{-1}$). This value was also similar to 300 g C m$^{-2}$ yr$^{-1}$

CAR reported by DeLaune and White[25] for LA coastal wetlands. However, the big difference between this study and Ouyang and Lee[24] is the areal data for the West Coast. Ouyang and Lee[24] reported 2685 km² tidal marsh, which is over four times larger than the area (640 km²) in this study. Our areal data are similar to 567.5 km² tidal wetlands in West Coast reported by Hinson et al.[16] who also extract the tidal wetland area from US wetland inventory dataset. Ouyang and Lee[24] did not provide details on how they obtained this data, but from their Fig. 1, the 2685 km² should be for the Northeast Pacific, which includes the tidal wetlands in the Pacific coast, Alaska, Mexico, and Canada. CEC[18] also reported the Northeast Pacific coastal area to have over 3423

km² tidal wetlands, with ~ 80% located in Mexico as mangroves and tidal shrub. It thus is highly possible that Ouyang and Lee[24] largely overestimated the tidal marsh area in the West Coast of the conterminous United States. Furthermore, we had 47 sites in the West Coast which was nearly six times higher than the number reported by Ouyang and Lee[24]. Therefore, our estimation based on these detailed soil CAR sites and NWI tidal wetlands database is more site-specific than previous reports.

Another advantage to our compiled soil C accumulation data is that we upscaled the site-specific CAR to the spatial area with NWI tidal wetland database. This method ensures that records from regions with high or low density of observations are not unduly weighed in the upscaling process. The EPA[20] estimated that 3.3 Tg yr⁻¹ C had accumulated annually in the top 1 m sediments of 29,000 km² coastal wetlands in the conterminous United States since 1996. However, there are no details on how the value is calculated. Owing to the significant difference in regional CAR, simply upscaling one single average C accumulation rate to national and/or regional levels would cause large uncertainties in the total C accumulation. For example, the lower Mississippi region has 8193 km² tidal wetlands with the average CAR at 271.9 g C m⁻² yr⁻¹. There will be 40% underestimation if we use the national CAR average rate 161.8 g C m⁻² yr⁻¹ for the regional upscaling. Similarly, upscaling according to mean values on the vegetation types would also underestimate the national C sequestration rate. In this study, the average CAR for tidal freshwater wetlands, brackish, salt marsh, and mangroves are 166.2, 179.7, 154.3, and 151.5 g C m⁻² yr⁻¹, respectively. By multiplying these rates with their corresponding areas, there is 4.19 Tg C accumulated annually in the conterminous United States, which is still 0.4 Tg C lower than the data estimated based on the complete NWI tidal wetlands sum (4.59 Tg C yr⁻¹).

Although our average CAR (161.8 g C m⁻² yr⁻¹) is much lower than the global mean value (210 ± 20 g C m⁻² yr⁻¹)[13], it is still nearly three times higher than the estimation (60 g C m⁻² yr⁻¹) by Hinson et al.[16] for all tidal wetlands in the conterminous United States. Hinson et al.[16] got 60 g C m⁻² yr⁻¹ CAR by assuming that the SAR for US coastal wetlands to be the same to sea level rise rate

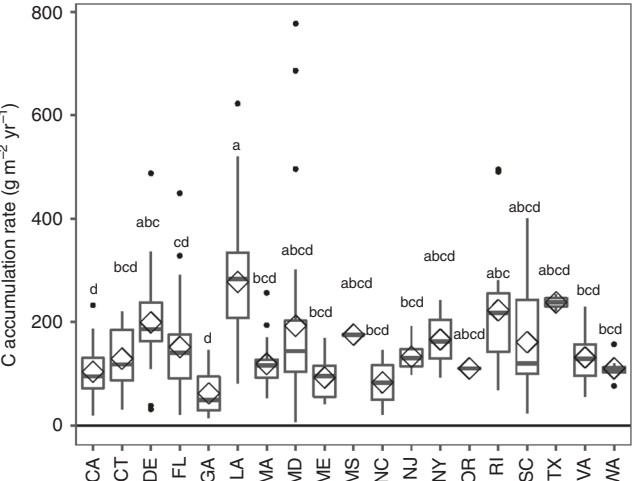

**Fig. 4** The boxplot of state level tidal wetlands C accumulation rates in the conterminous United States. Diamond is the mean value, the box is interquartile range, error bar is the largest and smallest value within 1.5 times interquartile range above 75% and below 25%, respectively. Black points indicate outside values. The different lower letters above each bar indicated significantly difference among states (Tukey HSD).

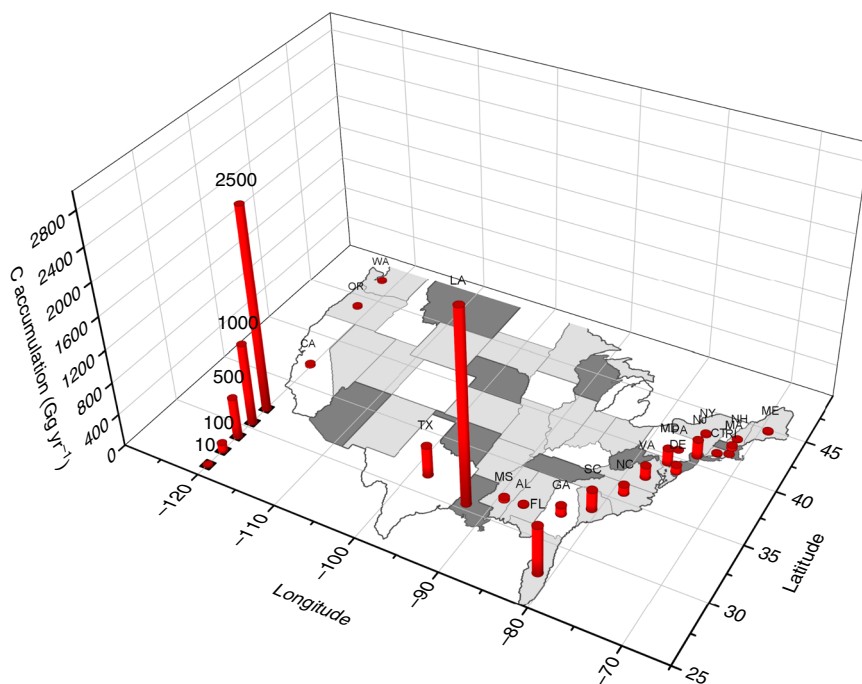

**Fig. 5** State level coastal wetland C accumulation (Gg yr⁻¹) in the tidal wetlands of the conterminous United States.

**Table 2 Conterminous United States tidal wetland sediment characteristics and area among different vegetation types (mean ± s.e.m.).**

| Vegetations | CAR $(g\ C\ m^{-2}\ yr^{-1})$ | C Density $(g\ C\ cm^{-3})$ | SAR $(mm\ yr^{-1})$ | Organic Sed. Rate $(g\ m^{-2}\ yr^{-1})$ | Mineral Sed. Rate $(g\ m^{-2}\ yr^{-1})$ | Area $(km^2)$ |
|---|---|---|---|---|---|---|
| Tidal Freshwater | 166.2 ± 15 | 0.038 ± 0.003[ab] | 6.13 ± 0.63[a] | 311 ± 37 | 950 ± 206 | 4828.8 |
| N | 44 | 44 | 41 | 41 | 44 | |
| Brackish | 179.7 ± 15 | 0.042 ± 0.002[a] | 4.70 ± 0.34[ab] | 374 ± 29 | 1239 ± 125 | 5233.5 |
| N | 81 | 86 | 80 | 81 | 81 | |
| Salt marsh | 154.3 ± 8 | 0.035 ± 0.001[b] | 4.42 ± 0.22[b] | 323 ± 19 | 1092 ± 94 | 13157.3 |
| N | 157 | 175 | 197 | 168 | 168 | |
| Mangrove | 151.5 ± 16 | 0.046 ± 0.002[a] | 4.56 ± 0.58[ab] | 324 ± 36 | 586 ± 119 | 2672.4 |
| N | 28 | 28 | 25 | 28 | 26 | |
| p values | ns | <0.05 | <0.05 | ns | ns | |

*SAR* sediment accretion rate, *CAR* carbon accumulation rate, *BD* bulk density, *N* number of observations. Different lowercase letters indicated a significant difference among vegetations (Tukey HSD)

**Table 3 Correlation coefficient ($r$) of tidal wetlands sediment variables with regional and climate factors.**

| Variables | SAR | C density | RSLR | Salinity | Tair | Prcp | Latitude | Longitude |
|---|---|---|---|---|---|---|---|---|
| CAR | 0.78** | 0.55** | 0.40** | −0.15* | 0.18** | 0.19** | −0.18** | 0.11* |
| SAR | | −0.18** | 0.48** | −0.23** | 0.32** | 0.11 ns | −0.33** | −0.10 ns |
| C density | | | −0.02 ns | −0.19* | −0.04 ns | −0.04 ns | 0.05 ns | 0.18** |

*$p < 0.05$. **$p < 0.01$, ns: not significant. The correlations among CAR, SAR and C density were partial correlations with C density and SAR as the control, respectively

($2\ mm\ yr^{-1}$) and C density to be $0.03\ g\ cm^{-3}$. As a result of this extrapolation, Hinson et al.[16] estimated that there was only ~1.5 Tg C sequestrated by tidal wetlands annually. Although their estimation of sea level rise rate is reasonable, the SAR was reported to keep pace with RSLR rather than sea level rise rate[10], because RSLR is a combination of sea level rise and vertical land movements[26]. In Lower Mississippi region, for example, land subsidence rate is 3–4 times faster than sea level rise rate (~$1.8\ mm\ yr^{-1}$ for the global mean in the last 50 yrs), and RSLR reached to $8.82\ mm\ yr^{-1}$. In this study, the average SAR rate ranged from $3.2\ mm\ yr^{-1}$ (Pacific Northwest) to $8.9\ mm\ yr^{-1}$ (Lower Mississippi), both of these two values were much higher than the SAR rate assumed by Hinson et al.[16]. Moreover, as mentioned above, simple upscaling using average national data would greatly underestimate the C sequestration amount in regions with high CAR, like the Lower Mississippi, which has large areal tidal wetlands.

Our results showed that vegetation types had no significant effect on CAR. Chmura et al.[13] also found no significant difference in CAR between salt marsh and mangroves from a global dataset. Although there was no CAR difference among vegetation, the C density and SAR results still suggest that underlying mechanisms that regulated CAR may vary among different vegetation types. In this study, mangroves had significantly higher C density ($0.046 ± 0.002\ g\ cm^{-3}$) than salt marsh ($0.035 ± 0.001\ g\ cm^{-3}$). Chmura et al.[13] also reported that the average soil carbon density in mangrove ($0.055 ± 0.004\ g\ cm^{-3}$) was much higher than that in salt marsh ($0.039 ± 0.003\ g\ cm^{-3}$) across the global data set. Tidal freshwater wetlands contained the highest SAR but the lowest C density, whereas mangroves had the highest C density but lower SAR (Table 1). The result thus suggests that SAR plays a more important role than C density in tidal freshwater wetlands to build CAR, whereas C density would have a greater contribution to CAR in mangroves.

Vertical SAR is important to tidal wetlands C sequestration. From the pathway analysis, RSLR was found to be the controlling factor with a significant effect on tidal wetland CAR, mainly through SAR and organic sedimentation. From the compiled 343 sediment cores analyses, the SAR observed in most regions of the conterminous United States have capacity to keep pace with RSLR (Table 2). Geomorphic models[7,27–29] showed enhancement in CAR under a modest increase in RSLR, whereas rapid C loss or reduction of CAR occurred when the SAR surpassed its optimal point. Morris et al.[7] predicted that the limiting rate of RSLR on the southeast coastal marshes of US was $12\ mm\ yr^{-1}$. This rate was much higher than the current RSLR in most regions except the Lower Mississippi where RSLR reached $8.82\ mm\ yr^{-1}$ owing to rapid subsidence[25]. Therefore, tidal wetlands in Lower Mississippi may have reached or in some cases even surpassed their threshold of sustainability[30], which could attribute to the current rapid area loss, as already indicated in previous studies[18,20,25,31].

The resilience of tidal wetlands to sea level rise depends on its vertical accretion rate and/or potential horizontal migration to upland. With ongoing climate change, the CAR in tidal wetlands can be either maintained or even increased as long as SAR keeps pace with RSLR[9,10,32]. Important to these processes is the relation with the long-term SAR, which relies on the sedimentation of mineral and organic matter on the marsh surface[33]. One of the important drivers in these processes is that the increases in tidal inundation, which have been shown to promote more frequent and longer duration of mineral sediment settling on the wetland platform, faster vegetation growth, and more organic matter accumulation[7,29]. Thus, the ecogeomorphic feedbacks tend to maintain the SAR at a similar pace as that of the RSLR. In the pathway analysis (see results section for details), RSLR is shown to have a significant effect on organic sedimentation rather than on mineral sedimentation, indicating that plant originated organic sedimentation feedbacks to RSLR drives the resilience of tidal wetland vertical accretion to sea level rise. This result was also in agreement with Morris et al.[28] that organic sedimentation contributes ~60% to vertical accretion in the East Coast estuaries.

Horizontal migration also helps tidal wetlands to adapt and survive under sea level rise. Wetland losses at the seaward margin could be offset by the migration to highland[27,34], and the net changes in tidal wetland area would be minimal or even increase.

However, human activities greatly affect the landward migration of tidal wetlands in costal regions. Schuerch et al.[23] predicted global tidal wetland area change based on an integrated modeling approach, which considered both the ability of coastal wetlands to build up vertically and laterally by sediment accretion and the accommodation space, respectively. They found that global tidal wetland would increase ~60% of the current area if these wetlands have sufficient accommodation space, but could lose up to 30% of their aerial extent assuming no further accommodation space in addition to current levels[23].

Our projection of the conterminous United States tidal wetlands C sequestration considered both wetland area changes (extracted from the model by Schuerch et al.[23]) and CAR response to future RSLR. Under the most-restricted accommodation space status (Supplementary Fig. 1, no accommodation space for population density higher than five people km$^{-2}$), the conterminous United States tidal wetlands area in 2100 will lose up to 25% of their aerial extent in 2100, but the net CAR loss would be <10 %. Similarly, the tidal wetlands net annual C gains would be doubled but the net area only increased by <50% under the upper estimates for current accommodation space in 2100 (Fig. 7). Our data indicated that annual C sequestrated by tidal wetlands in the conterminous United States would remain consistent until at least to 2100 under RCP 2.6, RCP 4.5, and RCP 8.5 scenarios even under the most-restricted accommodation space. These results highlight the ecogeomorphic feedbacks between RSLR and C accumulation in tidal wetlands, which has been well established in some recent studies[9,10,27,35].

The national extrapolation and prediction of our observed patterns are intended to contextualize the empirical relationship between RSLR and CAR. Unlike process-based models, the empirical relationships observed here cannot capture the complex processes that regulate the long-term C accumulation in coastal wetlands[7]. We stress that our prediction of the C sequestration rate in regional and national tidal wetlands of the conterminous United States to 2100 would underestimate the future uncertainties. In this study, future CAR prediction is based on the mean values of modeled RSLR by Kopp et al.[22], but did not consider the distribution of these modeled RSLR data. The inherited weakness in this linear extrapolation thus would underestimate the uncertainty of future CAR, and reduce its predictive capability. However, the value of these linear approximations lie in their descriptive strength rather than predictive

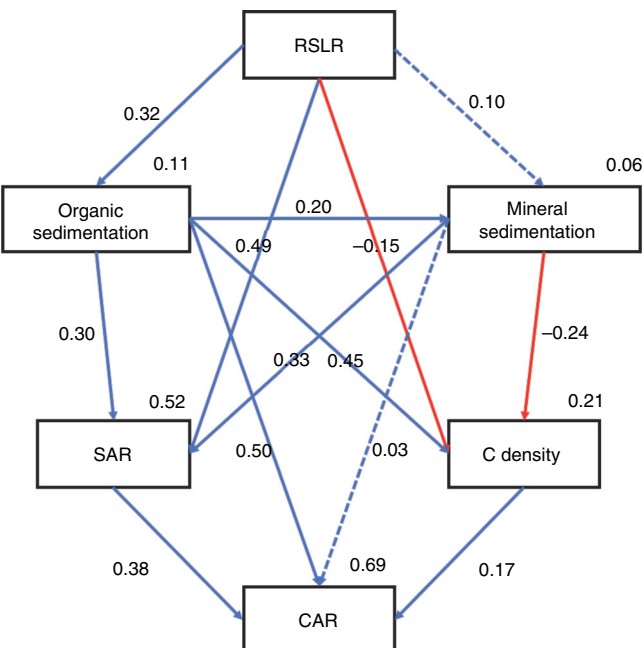

**Fig. 6** Pathway analysis is showing the direct and indirect effect of RSLR on CAR. RMSEA = 0.046, GFI = 0.996, $\chi^2$ = 3.12, Probability = 20.5%. Solid (Blue: Positive; Red: Negative) arrows indicate significant ($p < 0.05$) effects. Values associated with solid arrows represent standardized path coefficients. $R^2$ values associated with response variables indicate the proportion of variation explained by relationships with other variables. The high probability associated with the $\chi^2$ tests indicate good model fit to data, i.e., no significant discrepancies.

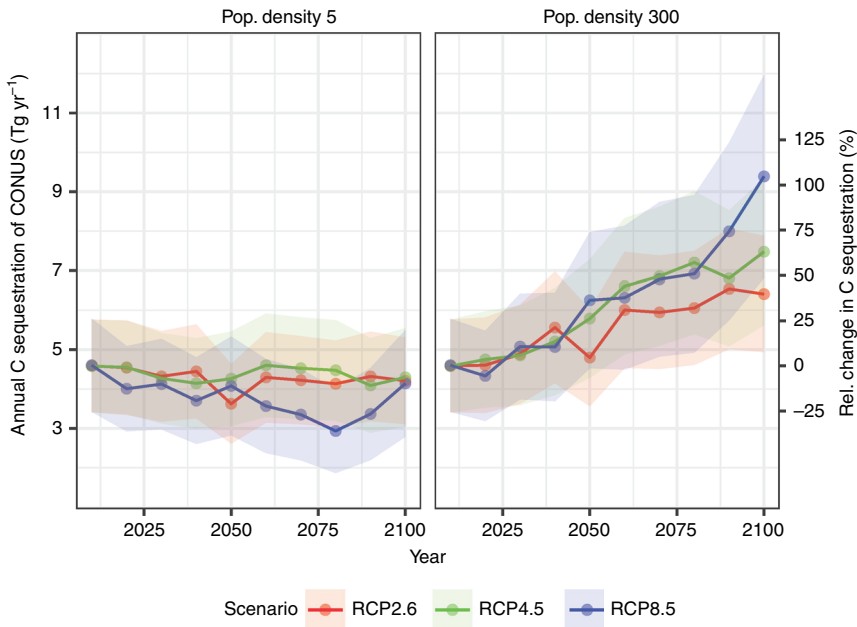

**Fig. 7** Estimation of total annual C sequestration of the conterminous United States tidal wetlands to 2100 based on tidal wetland area changes estimated by and projected CAR based on future RSLR under RCP 2.6, RCP 4.5, and RCP 8.5 scenarios. The colored shadow is the probability distribution between 2.5th and 97.5th percentiles under each scenario.

capability[36]. This study described the RSLR-sensitivity of C accumulation rate in tidal wetlands, and can serve as an guideline for other process-based models.

The predicted increases in CAR with RSLR along the conterminous United States as outlined here are in agreement with studies that examined historical records[35]. For instance, Rogers et al.[35] reported that tidal marshes experienced rapid RSLR over the past few millennia and had 1.7–3.7 times higher soil C concentration than those subject to long-term sea level stability. As highlighted in our projections, the higher RSLR creates more vertical and lateral accommodation space for C storage in tidal wetlands, which allows these wetlands to survive and increase their C sequestration under future climate change scenarios. In conclusion, this national upscaling of tidal wetlands CAR data allow us to qualify the importance of these ecosystems in climate mitigation, which may be used to assist in mitigating the $CO_2$ emission if they are maintained and restored in the conterminous United States. Our analysis provides empirical data to support the future role of tidal wetlands C sequestration potential as a result of their resilences to sea level rise, which can be maintained or even increased in most regions of the conterminous United States by the end of this century under all IPCC scenarios. The results presented here can serve as a baseline assessment of C sequestration in tidal wetlands of the conterminous United States and suggests that conservation and restoration of tidal wetlands could be a great benefit in reducing atmospheric carbon even under the more severe climate change scenarios.

## Methods

**Data collection.** We examined 64 published studies (please see references in the Supplementary Information) that reported sediment accretion rates and soil C density or parameters necessary for estimating C density (BD, soil organic matter content, or soil C content) in tidal wetlands of the conterminous United States. From these literatures, we compiled 372 data sites, of which 310 sites reported C accumulation rate (CAR), or in which CAR can be calculated based on reported values; and of which 343 sites reported SAR (Supplementary Table 1). The detailed location of these sites was extracted from the primary literature by using Google Maps.

For most of these studies, soil C density was not reported. Soil C content data in some studies were derived from the measurement of loss on ignition (LOI). LOI measurement of mangrove and tidal freshwater marsh soils were transformed into organic C content by dividing a factor of 1.724[37]. For brackish and salt marsh soils, we applied the quadratic relationship specific to salt marshes reported by Craft et al.[38]: $TOC = 0.04 \times LOI + 0.0025 \times LOI^2$.

BD was also not reported in some sites. The missing BD was calculated based on a mixing model which describes the BD as a function of LOI in intertidal wetland sediments[28]. The model assumes that the bulk volume of sediment is equal to the sum of self-packing volumes of organic and mineral components or $[BD = 1/[LOI/k_1 + (1 - LOI)/k_2]$, where $k_1$ and $k_2$ are the self-packing densities of the pure organic and inorganic components, respectively. The values of $k_1$ and $k_2$ were estimated to be $0.085 \pm 0.0007 \, g \, cm^{-3}$ and $1.99 \pm 0.028 \, g \, cm^{-3}$, respectively[28].

Most of the collected studies reported the vertical SAR in recent decades, which allowed us to calculate the carbon accumulation rates (CAR). The vertical SAR represented average SARs from decades to centuries, depending on the different dating methods. The $^{137}Cs$ and $^{210}Pb$ dating methods were employed in 317 sites to determine decadal rates of vertical accretion, and 24 sites measured sub-decadal and decadal SAR based on the surface elevation table methods. Only one study used $^{14}C$ dating method to rebuild centuries to thousands of years accretion rates[11]. The $^{14}C$ dating method usually has a lower SAR than the method using $^{137}Cs$ and $^{210}Pb$[39]. To avoid statistical skew by dating methods, this study was not included in the final CAR calculation. Where reports made available both SAR and C density, the CAR was calculated. Some studies have reported the SAR and C density or C content for multiple layers, reflecting their changes over time. For these studies, we averaged their C density and SAR over the up 30 cm soils, which recorded the most recent C accumulation (<100 yrs).

Besides the soil C-related parameters, other environmental variables, i.e., mean annual temperature (Tair), mean annual precipitation (Prcp), the RSLR, and salinity were also collected. Tair and Prcp for each site were acquired from National Centers for Environmental Prediction atmospheric reanalysis[40]. RSLR data were collected from the Permanent Service for Mean Sea Level database[41]. Annual tide-gauge data were smoothed by fitting a temporal linear model to calculate the RSLR. To avoid the recorded RSLR rate being dominated by decadal variability in different tidal gauges[42,43], we standardized the long-term tidal gauges data set and only reported the most recent 60 yrs RSLR (from 1950s to 2010s, Supplementary

Table 3). To incorporate information from a broader set of tide-gauge records, we also included the tidal gauges, which have at least recent 30 yrs continuous records. We established a linear relationship between recent 30 yrs RSLR and recent 60 yrs RSLR (see Supplementary Fig. 2) based on long-term tidal gauges records, and applied the linear model to these short-term tidal gauges data to rebuild their recent 60 yrs RSLR. We would like to point out that extending the 30 yr records using this linear relationship may not fit all stations, as some sites may have differing patterns. The RSLR data used in the complied studies was extracted from the nearest tidal gauge station RSLR data set. Salinity was also collected from these compiled studies.

All the reported coastal tidal wetlands were categorized into four groups according to their salinity and dominant vegetation types. Tidal freshwater wetlands have salinity in the range from 0 to 0.5‰, brackish is the tidal marsh whose salinity ranged from 0.5 to 18‰, and salt marshes have the salinity over 18‰. In salt marsh, many studies also reported the difference between high marsh and low marsh, o to their distinguished elevation distribution.

Locations and vegetation types of tidal wetlands were extracted from the National Wetland Inventory (NWI) data set (https://www.fws.gov/wetlands/). The NWI uses a classification system for aquatic habitats, including systems, subsystems, and classes. The wetlands extracted from the NWI for this study contain not only 'blue carbon' in the IPCC Wetlands Supplement[44], but also tidal freshwater wetlands. Based on their locations and NWI classifications, we extracted 325,255 tidal wetland polygons from the NWI database and classified them into four types: tidal freshwater wetlands, brackish, salt marsh, and mangroves. The detailed classification codes of the four vegetations were listed in Supplementary Table 2. We further assigned state name and watershed region (HUC2 polygons from the Watershed Boundary Dataset: https://www.usgs.gov/core-science-systems/ngp/national-hydrography/watershed-boundary-dataset) to each tidal wetland. CAR, SAR, C density, Tair, Prcp, and RSLR, were also assigned to these tidal wetlands.

**Extrapolation.** The geostatistical principle assumes that vegetation distribution gradually changes with environmental factors, like latitude, longitude, RSLR, and salinity (Fang et al. 2012; Reich et al. 2014). We thus assumed that C accumulation at one sampling site might have the highest similarity to that at the nearby sites. Spatial interpolation methods have been used in this study to calculate the CAR of each of the tidal wetlands in the conterminous United States: the CAR of each tidal wetland polygon (totally 325,255) was estimated based on compiled sites within the radius of 100 km. The 95% confidence intervals (CI) of each tidal wetland polygon was also estimated by assuming that nearby available CAR data sites could represent the CAR values for the wetland. There were still many wetlands (totally 151,962) that do not have any or only one nearby available value. For these wetlands, we used the regional mean and CI values to estimate their CAR and CI. The state level and regional C sequestration amounts were summarized based on the data for each tidal wetland. The CI of summed values was propagated based on the below equations (E1 & E2)[45]:

$$\delta_{sum} = \sqrt[2]{\sum_{i=1}^{n} \delta i^2} \qquad (1)$$

$$95\% CI = 1.96 \times \delta_{sum} \qquad (2)$$

where $\delta_{sum}$ is the standard error of the sum of wetlands, $\delta_i$ is the standard error of each wetland. CI is the 95% confidence intervals.

**Statistical analysis.** The CAR, SAR, C density, organic and mineral sedimentation data were grouped by vegetation types, HUC2 watershed regions, and states. We separated all the US conterminous data into seven watershed regions: New England, Mid Atlantic region, South Atlantic and Gulf, Lower Mississippi, Texas-Gulf, California, and Pacific-North (Fig. 1). For salt marsh data, we further compared their difference between low marsh and high marsh.

One-way analysis of variance (ANOVA) was used to detect the difference in above variables among vegetations and watershed regions. The CAR changes among all coastal States were also analyzed with one-way ANOVA. Turkey-HSD was then conducted for multiple comparisons. The ANOVA was performed by using the aov function in the base package of R version 3.3.2 (R Development Core Team, 2016).

Correlation analysis and pathway analysis were conducted to investigate the relationship between tidal wetland CAR and climate and environmental variables. Partial correlation was conducted for the analysis among CAR, SAR, and C density. The pathway analysis was implemented using the maximum likelihood estimation method and was fitted with the $\chi^2$ test. The Pearson correlation and partial correlation was performed using R version 3.3.2. Pathway analyses were conducted using the Amos 21.0 (IBM SPSS Inc, Chicago, IL, USA).

To select the most relevant predictors of CAR, we used stepwise algorithm based on AIC selection with linear mixed models. The independent variable was CAR, whereas the predictors were HUC2 watershed regions, vegetation types, MAT, MAP, RSLR, Longitude, Latitude. That is, CAR ~ HUC2 + Vegetation + MAT + MAP + RSLR + Longitude + Latitude, with the source of reference as random factor. Only RSLR was included in the most parsimonious final model. We

thus used the mixed linear relationship between CAR and RSLR to predict CAR with projected RSLR data[22] in IPCC Representative Concentration Pathway (RCP) scenarios.

Prediction of the conterminous United States tidal wetland C accumulation: The spatial and temporal prediction of the conterminous United States tidal wetlands C accumulation was based on the projected tidal wetland area changes by Schuerch et al.[23] under different IPCC scenarios and population density threshold. In brief, Schuerch et al.[23] predicted global tidal wetland area change based on 12,148 coastal line segments from the Dynamic Interactive Vulnerability Assessment (DIVA) modeling framework[46] by a integrated modeling approach, which considered both the ability of coastal wetlands to build up vertically and laterally by sediment accretion and the accommodation space, respectively. They set four population density thresholds (i.e., 5, 20,150, and 300 people per km$^2$) for lateral accommodation space availability, which means the segments have higher population density than the thresholds would have no lateral accommodation space for the tidal wetland. In this study, we extract the conterminous United States tidal wetland area percentage changes from the global 12,148 coastal line segments under all population thresholds in RCP 2.6, RCP 4.5, and RCP 8.0 scenarios.

To estimate changes in future CAR under projected IPCC scenarios by 2100, we applied the mixed linear model predicted CAR by RSLR. We extracted the conterminous United States future RSLR mean values from Kopp et al.[22] at decadal intervals for locations of each coastal wetland segment under RCP 2.6, RCP 4.5, and RCP 8.5 trajectories. Site-to-site differences in the RSLR projections[22] originate from varying non-climatic background uplift or subsidence, oceanographic effects, and spatially variable responses of the geoid and the lithosphere to shrinking land ice. The projected CAR was thus calculated based on RSLR differences between current (the 2010s) values and future values. However, CAR generally reached a maximum rate if SAR cannot keep pace with RSLR[29]. Previous study predicted a critical RSLR rate of about 10 mm yr$^{-1}$ for many typical salt marshes in the US and Europe[8]. Below the critical RSLR rate, tidal wetlands are stable ecosystem, and CAR will keep increase with higher RSLR till the critical rate reached. However, this 10 mm yr$^{-1}$ critical rate may lead to a conservative estimation of CAR in response to future RSLR in many US coastal wetlands as Morris et al.[7] predicted that the critical rate of RSLR on the southeast coastal marshes of US was 12 mm yr$^{-1}$. The projected coastal wetland C accumulation amount till 2100 was calculated based on projected future CAR and coastal wetland area changes for each coastline segment.

**Reporting summary**. Further information on research design is available in the Nature Research Reporting Summary linked to this article.

## Data availability

A Source Data File, containing the raw data underlying the research and all figures presented in our paper, is available in the Supplementary Information. Figure 1 was extracted from Natioanl Wetland Inventory database (https://www.fws.gov/wetlands/) based on the wetland type code attached in the Supplementary Table 2. The predicted global wetland area change data were extracted from the model by Schuerch et al. 2018 (https://gitlab.com/mark.schuerch/global-coastal-wetland-model.git.). Correspondence and requests should be addressed to F.W.

## Code availability

The code used in this study is available from F.W. on request.

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

## Acknowledgements

This study was partially funded by Natural Science Foundation of China (31300419, 31670621, 31870463), the Key Special Project for Introduced Talents Team of Southern Marine Science and Engineering Guangdong Laboratory (Guangzhou) (GML2019ZD0408), R&D Program of Guangdong Provincial Department of Science and Technology (2018B030324003) and Pearl River Nova Program of Guangzhou (201710010140) awarded to F.W. J.T. is supported by the NOAA National Estuarine Research Reserve Science Collaborative (NA09NOS4190153 and NA14NOS4190145). C.J.S. is supported by Australian Research Council (DE160100443).

## Author contributions

F.W. conceived and designed the study, collected and analyzed the data, and draft the first version of paper. X.L. performed GIS analysis. All authors contributed to the writing and editing of the orginal manuscript. F.W., C.S., and J.T. contributed to the revision process of the manuscript.

## Competing interests

The authors declare no competing interests.
