## [Peer Review File · Nature Communications]

Reviewers' comments:

Reviewer #1 (Remarks to the Author):

Tidal wetland resilience to sea level rise increases their carbon sequestration capacity in the conterminous United States

The manuscript addressed a topic of international importance to merit publication within a high profile journal such as Nature Communications. Tidal wetlands are among the most valuable ecosystems on Earth, and coasts are disproportionately vulnerable to global change.

There are, however, several significant limitations to the manuscript, which must be addressed. These will require some time consuming work. I thought the manuscript was very descriptive with a lack of a suitable discussion. The methodology was unclear. I found the figures to be interesting but the links to the text were vague.

Regarding my main points, I have kept my review succinct and just concentrated on one aspect of the paper.

1. I am concerned with how the authors calculated the relative sea level rise for each site.

a. Where is the methodology to calculate rates of relative sea-level rise? Where did they get the data from?

b. For a paper of this magnitude they should take the original data from NOAA or PMSL and standardize it so it has the same start or end date. They must make sure the instrumental data is at least 60 years old to avoid the rates being dominated by decadal variability. At the moment I am concerned the authors are comparing rates of different record lengths.

c. I am concerned the authors are unaware the difference between relative sea level and global mean sea level. Although the authors are correct that the high rates of relative sea-level are greater in the Mississippi delta than anywhere else in the US because of sediment compaction causing subsidence, there is no mention of the other processes that cause relative sea-level to differ from the global mean (glacial isostatic adjustment, tectonics, ocean dynamics, static equilibrium, etc.).

d. How is the depth of soil the authors use to calculate carbon storage, accretion rates, etc. related to the length of the sea-level record. For example does the depth of soil approximate the last 100 years but the RSL is much shorter? Related to this, should the author be considering a lag between the rate of sea-level rise and the coastal wetland response?

2. The take home points of this paper is the future of coastal wetland under increasing rates of sea-level rise. But I am concerned that the authors took the easy solution of apply the IPCC global mean sea level projections.

a. My concern of the difference between global mean and relative sea level is born from the projections the authors use...why are they using global mean projections when they illustrate that the Mississippi delta is already rising 3x the average rate in the 20th century?

b. The authors must use regional projections of relative sea-level. There are many papers published on this and differing states have their regional projections. But probably the easiest for the authors to use would be Kopp et al., 2014

Kopp, R. E. et al. Probabilistic 21st and 22nd century sea-level projections at a global network of tide-gauge sites. *Earth's Future* 2, EF000239 (2014).

This paper provide an aggregation of the individual components of relative sea-level change (Kopp et al. describes this as local sea level) for each tide gauge in the US. They project three ice sheet components (the Greenland Ice Sheet, West Antarctic ice sheet, and the East Antarctic ice sheet); glacier and ice cap surface mass balance; global mean thermal expansion and regional ocean steric and ocean dynamic effects; land water storage; and long-term, local, non-climatic sea-level change due to factors such as glacial isostatic adjustment, sediment compaction, and tectonics.

I also suggest the authors examine the differences between RCP 2.6 and 8.5. The paper by Horton

et al. (2018) clearly illustrated the significant different of emission scenarios for the resilience of coastal wetlands

Horton, B.P., Shennan, I., Bradley, S.L., Cahill, N., Kirwan, M., Kopp, R.E., Shaw, T.A., 2018. Predicting marsh vulnerability to sea-level rise using Holocene relative sea-level data. Nature Communications. DOI: 10.1038/s41467-018-05080-0.

Applying these new projections will significantly influence the results of the paper

Reviewer #2 (Remarks to the Author):

This study uses existing data to compile carbon accumulation rates in tidal wetlands in the U.S. Sites were categorized by dominant vegetation and grouped large watershed regions. Sediment and carbon accumulation rates were compared to estimated rates of sea level rise using IPCC scenarios. This is a useful synthesis of carbon accretion in coastal systems, a topic which is of great interest to those working in coastal ecosystems climate mitigation. My main concern is that the paper needs to be more explicit about the methods used in the original papers, the sources of error in the estimates and the implication of not having data on key variables related to accretion rates and SLR, such as elevation. Site elevation can have a large effect on accretion rates and since this data is available, it is curious why it wasn't used. What impact might this have on the results?

Specific comments

Line 29 – what is the variability associated with this estimate? A measure of variability is needed to inform readers of the uncertainty.

Line 97 – I would argue that there is a lot known about the processes that control C sequestration (per many of the references cited), this might better say that the balance of the processes and their relative contribution to C sequestration is not known.

Line 133 – some discussion of the fact that TX had only 2 sites on which to base this estimate is warranted.

Line 260 – Scenarios not sceneries

Lines 263 – there is some repetition here with earlier text – could be more concise

Line 270 – this appears to be a hypothesis based on the path analysis and not a conclusion based on the data (“dominated the resilience of tidal wetland vertical accretion”). I would reframe this comment.

Line 279 - this is a bit repetitive.

312 – seems to be a word missing here.

Line 361 – the methods used to make the SAR measures are important to the results. Saying that the 137-Cs and 210-Pb methods were “usually employed” isn't complete. How many estimates were made this way? For those that weren't, what other methods were used? This is critical to interpreting the results, particularly if marker horizons were used in some studies.

Figure 5 is difficult to read, perhaps a different format for these data.

Responses to Reviewers' comments:

Reviewer #1 (Remarks to the Author):

Tidal wetland resilience to sea level rise increases their carbon sequestration capacity in the conterminous United States

The manuscript addressed a topic of international importance to merit publication within a high profile journal such as Nature Communications. Tidal wetlands are among the most valuable ecosystems on Earth, and coasts are disproportionately vulnerable to global change.

There are, however, several significant limitations to the manuscript, which must be addressed. These will require some time consuming work. I thought the manuscript was very descriptive with a lack of a suitable discussion. The methodology was unclear. I found the figures to be interesting but the links to the text were vague.

Regarding my main points, I have kept my review succinct and just concentrated on one aspect of the paper.

Response: We would like to thank the reviewer for the positive comments on this study. In this revised version, we modified the paper based on these and the other reviewers' comments. Below you will see our point by point response to these comments.

1. I am concerned with how the authors calculated the relative sea level rise for each site.
a. Where is the methodology to calculate rates of relative sea-level rise? Where did they get the data from?

Response: We extracted the RSLR data based on the NOAA version, which is available at: <https://tidesandcurrents.noaa.gov/sltrends/mslUSTrendsTable.html> . This table includes all the US station data, and the time range from 30 yrs to over 100 yrs along the the conterminous U.S . For clarity, we have updated our method section to take into account these concerns. See line 356: "RSLR was collected from the nearest National Oceanic and Atmospheric Administration (NOAA) tide gauge (<https://tidesandcurrents.noaa.gov/sltrends/mslUSTrendsTable.html>)."

b. For a paper of this magnitude they should take the original data from NOAA or PMSL and standardize it so it has the same start or end date. They must make sure the instrumental data is at least 60 years old to avoid the rates being dominated by decadal variability. At the moment I am concerned the authors are comparing rates of different record lengths.

Response: Thanks. The RSLR data was extracted from NOAA dataset in <https://tidesandcurrents.noaa.gov/sltrends/mslUSTrendsTable.html>. In the dataset, there are 109 stations located in the conterminous US, and over half of them (55 stations) have at least 60 yrs of instrumental records of sea level data. For the remaining 54 stations, the records ranged from 30 yrs to 60 yrs. Among the 109 stations, 92 of them have the same end year (2017). Since most of the station have continuous records of RSLR, at least 30 yrs, we believe that the RSLR data represent long-term trends of RSLR in each region.

Furthermore, the relationship between C accumulation and RSLR has been confirmed in a recent paper by Rogers et al (2019 Nature), who reported that tidal marshes experienced rapid RSLR over the past few millennia have 1.7 to 3.7 times higher soil C concentration than those subject to long term sea-level stability. The higher RSLR would create more vertical and lateral accommodation space for C storage in tidal wetlands, which allows these wetlands to survive and increase their C sequestration under future climate change scenarios. The observed patterns in this study confirmed this study and indicated that RSLR can be a good predictor of C accumulation. We now incorporate this information into the manuscript for clarity, line 306: "For instance, Rogers, Kelleway et al. (2019) reported that tidal marshes experienced rapid RSLR over the past few millennia and had 1.7 to 3.7 times higher soil C concentration than those subject to long term sea-level stability. As highlighted in our projections, the higher RSLR creates more vertical and lateral accommodation space for C storage in tidal wetlands, which allows these wetlands to survive and increase their C sequestration under future climate change scenarios."

c. I am concerned the authors are unaware the difference between relative sea level and global mean sea level. Although the authors are correct that the high rates of relative sea-level are greater in the Mississippi delta than anywhere else in the US because of

sediment compaction causing subsidence, there is no mention of the other processes that cause relative sea-level to differ from the global mean (glacial isostatic adjustment, tectonics, ocean dynamics, static equilibrium, etc.).

Response: Thanks and to avoid any confusion, In the revised version manuscript, we used the modeled mean RSLR data by Kopp et al. (2014), who reported global RSLR projections under different IPCC scenarios originated from varying non-climatic background uplift or subsidence, oceanographic effects, and spatially variable responses of the geoid and the lithosphere to shrinking land ice. Please see line 427-431.

However, we did not include the full distribution of these modeled RSLR data for CAR prediction, which may underestimate the uncertainty of our prediction. In this study, the national extrapolation and prediction of our observed patterns is intended to contextualize the empirical relationship between RSLR and CAR. Unlike process-based models, the empirical relationships observed here cannot capture the complex processes that regulate the long-term C accumulation in coastal wetlands. However, the value of these linear approximations lie in their descriptive strength rather than predictive capability (Crowther, Todd-Brown et al. 2016). Moreover, the predicted increases in CAR with RSLR along the conterminous US as outlined here are in agreement with studies that exam historical records(Rogers, Kelleway et al. 2019), which reported that tidal marshes experienced rapid RSLR over the past few millennia and had 1.7 to 3.7 times higher soil C concentration than those subject to long term sea-level stability. Therefore, our extrapolation and prediction in can serve as an guideline for other process-based models.

To address the potential concerns from reviewers and readers, we discussed these uncertainties in line 294-311: " The national extrapolation and prediction of our observed patterns is intended to contextualize the empirical relationship between RSLR and CAR. Unlike process-based models, the empirical relationships observed here cannot capture the complex processes that regulate the long-term C accumulation in coastal wetlands^(Morris, Sundareshwar et al. 2002). We stress that our prediction of the C sequestration rate in regional and national tidal wetlands of the conterminous US to 2100 would underestimate the future uncertainties. In this study, future CAR prediction is based on the mean values of modeled

RSLR by Kopp, Horton et al. (2014), but did not consider the distribution of these modeled RSLR data. The inherent weakness in this linear extrapolation thus would underestimate the uncertainty of future CAR, and reduce its predictive capability. However, the value of these linear approximations lie in their descriptive strength rather than predictive capability (Crowther, Todd-Brown et al. 2016). This study described the RSLR-sensitivity of C accumulation rate in tidal wetlands, and can serve as a guideline for other process-based models. "

d. How is the depth of soil the authors use to calculate carbon storage, accretion rates, etc. related to the length of the sea-level record. For example does the depth of soil approximate the last 100 years but the RSL is much shorter? Related to this, should the author be considering a lag between the rate of sea-level rise and the coastal wetland response?

Response: In this study, we only focused on the most recent several decades, approximately 100 yrs variation in CAR and SAR. In some studies referenced, if the data given is over 100 yrs, we only utilized the most recent 100 yrs dataset. We have revised our method section to make it clear. See line 343 : "The ^{137}Cs and ^{210}Pb dating methods were employed in 317 sites to determine decadal rates of vertical accretion, and 24 sites measured sub-decadal and decadal SAR based on the surface elevation table (SET) methods."

2. The take home points of this paper is the future of coastal wetland under increasing rates of sea-level rise. But I am concerned that the authors took the easy solution of apply the IPCC global mean sea level projections.

a. My concern of the difference between global mean and relative sea level is born from the projections the authors use...why are they using global mean projections when they illustrate that the Mississippi delta is already rising 3x the average rate in the 20th century?

Response: We appreciate this comment and in the revised version of this study we have greatly improved our prediction models. As we described above, we used the recent projection of RSLR by Kopp et al 2014. In the projection, they reported global RSLR under different IPCC scenarios originated from varying non-climatic background uplift or

subsidence, oceanographic effects, and spatially variable responses of the geoid and the lithosphere to shrinking land ice. Based on the updated RSLR data, we have re-run our model to make accurate CAR projections.

b. The authors must use regional projections of relative sea-level. There are many papers published on this and differing states have their regional projections. But probably the easiest for the authors to use would be Kopp et al., 2014

Kopp, R. E. et al. Probabilistic 21st and 22nd century sea-level projections at a global network of tide-gauge sites. *Earth's Future* 2, EF000239 (2014).

This paper provide an aggregation of the individual components of relative sea-level change (Kopp et al. describes this as local sea level) for each tide gauge in the US. They project three ice sheet components (the Greenland Ice Sheet, West Antarctic ice sheet, and the East Antarctic ice sheet); glacier and ice cap surface mass balance; global mean thermal expansion and regional ocean steric and ocean dynamic effects; land water storage; and long-term, local, non-climatic sea-level change due to factors such as glacial isostatic adjustment, sediment compaction, and tectonics.

Response: We agree and now use the dataset suggested and conducted the projection based on updated RSLR data, please see our method section, line 426: "To estimate changes in future CAR under projected IPCC scenarios by 2100, we applied the mixed linear model predicted CAR from the site specific RSLR. We extracted the future conterminous U.S.RSLR data from Kopp, Horton et al. (2014) at decadal intervals for locations of each coastal wetland segment under RCP 2.6, RCP4.5 and RCP 8.5 trajectories. Site-to-site differences in the RSLR projections (Kopp, Horton et al. 2014) originate from varying non-climatic background uplift or subsidence, oceanographic effects, and spatially variable responses of the geoid and the lithosphere to shrinking land ice. The projected CAR was thus calculated based on RSLR differences between current (2010s) values and future values. However, CAR generally reached a maximum rate if SAR could not keep pace with RSLR. In our dataset, SAR

is a linear function of RSLR, and there is a critical RSLR (9.09 mm/yr) point for the conterminous U.S.tidal wetlands when SAR equals the RSLR. We thus assumed that CAR will reach its maximum when RSLR is higher than the critical RSLR point, and after this point the SAR is deemed to no keep pace with RSLR. The projected the conterminous U.S.coastal wetland C accumulation amount till 2100 was then calculated based on projected future CAR and coastal wetland area changes for each coastline segment.”

I also suggest the authors examine the differences between RCP 2.6 and 8.5. The paper by Horton et al. (2018) clearly illustrated the significant different of emission scenarios for the resilience of coastal wetlands.

Response: In the revised version manuscript, we modeled the future conterminous U.S. tidal wetland C sequestration under RCP2.6, 4.5 and 8.5 scenarios. We have described the detailed difference between the three scenarios along the Results and Discussion sections. See line: 282-293“Our projection of the conterminous US tidal wetlands C sequestration considered both wetland area changes (extracted from the model by Schuerch, Spencer et al. (2018)) and CAR response to future RSLR. Under the most restricted accomodation space status (Fig S1, no accomodation space for population density higher than 5 people km⁻²), the conterminous US tidal wetlands area in 2100 will lose up to 25% of their aerial extent in 2100 , but the net CAR loss would be less than 10%. Similarly, the tidal wetlands net annual C gains would be over 100% but the net area only increased by less than 50% under the upper estimates for current accomodation space in 2100. Our data indicated that annual C sequestrated by tidal wetlands in the conterminous US would remain consistent until at least to 2100 under RCP 2.6, RCP 4.5 and RCP 8.5 scenarios which is based on the most restricted accomodation space. These results highlight the ecogeomorphic feedbacks between RSLR and C accumulation in tidal wetlands, which has been well established in some recent studies (Kirwan and Megonigal 2013, Kirwan, Temmerman et al. 2016, Kirwan, Walters et al. 2016, Rogers, Kelleway et al. 2019), however, until now not based at individual study sites along the conterminous US.”

Horton, B.P., Shennan, I., Bradley, S.L., Cahill, N., Kirwan, M., Kopp, R.E., Shaw, T.A., 2018.

Predicting marsh vulnerability to sea-level rise using Holocene relative sea-level data. Nature Communications. DOI: 10.1038/s41467-018-05080-0.

Applying these new projections will significantly influence the results of the paper

Response: Thanks for your suggestions. In the revised version, we re-run the model projections based on your suggestions. We believe that this version is greatly improved.

Reviewer #2 (Remarks to the Author):

This study uses existing data to compile carbon accumulation rates in tidal wetlands in the U.S. Sites were categorized by dominant vegetation and grouped large watershed regions. Sediment and carbon accumulation rates were compared to estimated rates of sea level rise using IPCC scenarios. This is a useful synthesis of carbon accretion in coastal systems, a topic which is of great interest to those working in coastal ecosystems climate mitigation. My main concern is that the paper needs to be more explicit about the methods used in the original papers, the sources of error in the estimates and the implication of not having data on key variables related to accretion rates and SLR, such as elevation. Site elevation can have a large effect on accretion rates and since this data is available, it is curious why it wasn't used. What impact might this have on the results?

Response: We agree with the reviewer that elevation is an important factor, however, most of the published reports do not have detailed elevation data. Since tidal wetlands only exists between tidal ranges, the elevation variation is relatively much smaller among sites.

Although most studies did not report elevation of sampling site, many salt marsh studies compared the differences between high marsh and low marsh, of which these differences likely originate from elevation of the marsh. We thus conducted a linear mixed model for the high marsh and low marsh data. The result indicated that there was no difference between high marsh and low marsh on CAR and organic sedimentation, but low marsh had significantly higher mineral sedimentation rates. This result was consistent with our findings that only RSLR greatly affect CAR or organic sedimentation rates. Mineral sedimentation generally from the sea water mineral deposition, it thus reasonable that low marsh, with

more frequently sea water inundation should have higher rate of mineral sedimentation rate than high marsh.

We have added this comparison in the result section. See line 144: " Due to the distinguished elevation distribution, salt marsh can be divided into high marsh and low marsh. We also compared the difference between high marsh and low marsh. There was no significant difference in CAR and organic sedimentation rate between high marsh and low marsh, but low marsh had significantly higher mineral sedimentation than high marsh ($p < 0.05$). "

Specific comments

Line 29 – what is the variability associated with this estimate? A measure of variability is needed to inform readers of the uncertainty.

Response: We revised the abstract section based on the new projections. Please see line 24: "The conterminous US tidal wetlands sequestered 4.2-5.0 Tg Cyr^{-1} based on different upscaling methods."

Line 97 – I would argue that there is a lot known about the processes that control C sequestration (per many of the references cited), this might better say that the balance of the processes and their relative contribution to C sequestration is not known.

Response: Revised as suggested. See line 90: "Furthermore, the balance of controlling factors and their relative contributions to C sequestration is not well known in the conterminous US tidal wetlands."

Line 133 – some discussion of the fact that TX had only 2 sites on which to base this estimate is warranted.

Response: Revised as " TX only contains two reported data (Fig. 2), however the average of these sites represents the second highest value ($237.8 \pm 16 \text{ g C m}^{-2} \text{ yr}^{-1}$) among the coastal states. "

Line 260 – Scenarios not sceneries

Response: Revised as suggested.

Lines 263 – there is some repetition here with earlier text – could be more concise

Response: We revised this section based on your and the other reviewer's comments

Line 270 – this appears to be a hypothesis based on the path analysis and not a conclusion based on the data (“dominated the resilience of tidal wetland vertical accretion”). I would reframe this comment.

Response: Revised as suggested

Line 279 - this is a bit repetitive.

Response: Revised as suggested

312 – seems to be a word missing here.

Response: Revised as suggested

Line 361 – the methods used to make the SAR measures are important to the results. Saying that the ^{137}Cs and ^{210}Pb methods were “usually employed” isn't complete. How many estimates were made this way? For those that weren't, what other methods were used? This is critical to interpreting the results, particularly if marker horizons were used in some studies.

Response: We have added the detailed method information for each reference the supporting materials, see dataS1. In total, the ^{137}Cs and ^{210}Pb dating methods were employed in 317 sites to determine long-term average rates of vertical accretion, and 24 sites measured SAR based on the surface elevation table (SET) methods. No reports were by marker horizons methods. Both SET and radiometric method engaged into the decadal scale SAR measurement, and the comparison between the two methods by Breithaupt, Smoak et al. (2018) indicated that they generally have consistent rate at decadal SAR measurements.

To address your concerns, we revised our method section, see line 326: " The ^{137}Cs and ^{210}Pb dating methods were employed in 317 sites to determine decadal rates of vertical accretion, and 24 sites measured sub-decadal and decadal SAR based on the surface elevation table (SET) methods. Only one study used ^{14}C dating method to rebuild centuries to thousands of years accretion rates (Redfield and Rubin 1962). The ^{14}C dating method

usually has lower SAR than the method using ^{137}Cs and ^{210}Pb (Orson and Howes 1992). To avoid the statistical skew by dating methods, this study was not included in the final CAR calculation. "

Figure 5 is difficult to read, perhaps a different format for these data.

Response: For clarity, we have added a table in the supporting materials, see Table S1, and have opted to keep this figure in the manuscript as we feel it is important to highlight the special variation.

Reference

- Breithaupt, J. L., J. M. Smoak, R. H. Byrne, M. N. Waters, R. P. Moyer and C. J. Sanders (2018). "Avoiding timescale bias in assessments of coastal wetland vertical change." Limnol Oceanogr **63**(Suppl 1): S477-S495.
- Crowther, T. W., K. E. O. Todd-Brown, C. W. Rowe, W. R. Wieder, J. C. Carey, M. B. Machmuller, B. L. Snoek, S. Fang, G. Zhou, S. D. Allison, J. M. Blair, S. D. Bridgham, A. J. Burton, Y. Carrillo, P. B. Reich, J. S. Clark, A. T. Classen, F. A. Dijkstra, B. Elberling, B. A. Emmett, M. Estiarte, S. D. Frey, J. Guo, J. Harte, L. Jiang, B. R. Johnson, G. Kröel-Dulay, K. S. Larsen, H. Laudon, J. M. Lavellee, Y. Luo, M. Lupascu, L. N. Ma, S. Marhan, A. Michelsen, J. Mohan, S. Niu, E. Pendall, J. Peñuelas, L. Pfeifer-Meister, C. Poll, S. Reinsch, L. L. Reynolds, I. K. Schmidt, S. Sistla, N. W. Sokol, P. H. Templer, K. K. Treseder, J. M. Welker and M. A. Bradford (2016). "Quantifying global soil carbon losses in response to warming." Nature **540**: 104.
- Kirwan, M. L. and J. P. Megonigal (2013). "Tidal wetland stability in the face of human impacts and sea-level rise." Nature **504**(7478): 53-60.
- Kirwan, M. L., S. Temmerman, E. E. Skeehan, G. R. Guntenspergen and S. Fagherazzi (2016). "Overestimation of marsh vulnerability to sea level rise." Nature Climate Change **6**(3): 253-260.
- Kirwan, M. L., D. C. Walters, W. G. Reay and J. A. Carr (2016). "Sea level driven marsh expansion in a coupled model of marsh erosion and migration." Geophysical Research Letters **43**(9): 4366-4373.
- Kopp, R. E., R. M. Horton, C. M. Little, J. X. Mitrovica, M. Oppenheimer, D. J. Rasmussen, B. H. Strauss and C. Tebaldi (2014). "Probabilistic 21st and 22nd century sea-level projections at a global network of tide-gauge sites." Earth's Future **2**(8): 383-406.
- Morris, J. T., P. V. Sundareshwar, C. T. Nietch, B. Kjerfve and D. R. Cahoon (2002). "Responses of Coastal Wetlands to Rising Sea Level." Ecology **83**(10): 2869-2877.
- Orson, R. A. and B. L. Howes (1992). "Salt Marsh development studies at Waquoit Bay, Massachusetts: Influence of geomorphology on long-term plant community structure." Estuarine, Coastal and Shelf Science **35**(5): 453-471.
- Redfield, A. C. and M. Rubin (1962). "Age of salt marsh peat and its relation to recent changes in sea levels at Barnstable, Massachusetts." Proceedings of the National Academy

of Sciences of the United States of America **48**(10): 1728-1735.

Rogers, K., J. J. Kelleway, N. Saintilan, J. P. Megonigal, J. B. Adams, J. R. Holmquist, M. Lu, L. Schile-Beers, A. Zawadzki, D. Mazumder and C. D. Woodroffe (2019). "Wetland carbon storage controlled by millennial-scale variation in relative sea-level rise." Nature **567**(7746): 91-95.

Schuerch, M., T. Spencer, S. Temmerman, M. L. Kirwan, C. Wolff, D. Lincke, C. J. McOwen, M. D. Pickering, R. Reef, A. T. Vafeidis, J. Hinkel, R. J. Nicholls and S. Brown (2018). "Future response of global coastal wetlands to sea-level rise." Nature **561**(7722): 231-234.

Reviewers' comments:

Reviewer #1 (Remarks to the Author):

In my first review of the manuscript I had a major concern that the authors were using a time series of relative sea level rise data of different lengths with different start and end dates. As far as I can tell the authors have chosen not to address this (although it would be relatively easy by either standardizing the data or working with a statistician to use a spatiotemporal model

Therefore, I am just repeating my initial concerns:

For a paper of this magnitude they should take the original data from NOAA or PMSL and standardize it so it has the same start or end date. They must make sure the instrumental data is at least 60 years old to avoid the rates being dominated by decadal variability. At the moment I am concerned the authors are comparing rates of different record lengths.

Reviewer #2 (Remarks to the Author):

I've read the revised manuscript and the authors response to the reviewers comments and am satisfied with the corrections the authors have made. Overall, the paper is much improved and I recommend it for publication.

Responses to comments

Reviewer #1 (Remarks to the Author):

In my first review of the manuscript I had a major concern that the authors were using a time series of relative sea level rise data of different lengths with different start and end dates. As far as I can tell the authors have chosen not to address this (although it would be relatively easy by either standardizing the data or working with a statistician to use a spatiotemporal model.

Therefore, I am just repeating my initial concerns:

For a paper of this magnitude they should take the original data from NOAA or PMSL and standardize it so it has the same start or end date. They must make sure the instrumental data is at least 60 years old to avoid the rates being dominated by decadal variability. At the moment I am concerned the authors are comparing rates of different record lengths.

Responses to Referee #1:

In the revised version, we updated the relative sea-level rise (RSLR) data based on your suggestion. Specifically, RSLR data was collected from the Permanent Service for Mean Sea Level (PSMSL or PMSL) database (Holgate, Matthews et al. 2013). Annual tide-gauge data were smoothed by fitting a temporal linear model to calculate the RSLR. To avoid the recorded RSLR rate being dominated by decadal variability in different tidal gauges, we standardized the long-term tidal gauges dataset and only reported the most recent 60-year RSLR (from 1950s to 2010s, Data file S3 from supporting materials). To incorporate information from a broader set of tide-gauge records, we also included the tidal gauges which have at least recent 30-year continuous records. We established a linear relationship between recent 30-year RSLR and recent 60-year RSLR (Fig. R2b) based on long-term tidal gauges records, and applied the linear model to these short-term tidal gauges data (30- to 60-year continuous records) to rebuild their recent 60-year RSLR (see Table S3).

The RSLR data used in the compiled studies thus was extracted from the nearest tidal gauge station RSLR dataset. See line 354-363.

Fig R2. The relationships among different time scale RSLR rates recorded by tidal gauges (continuous records over 60 yrs) along the US coastal lines. (a) The relationship between recent 60 yrs RSLR and long-term RSLR (over 60 yrs) and (b) the relationship between recent 60 yrs RSLR and recent 30 yrs RSLR rate. The solid black lines represent the line of $y=x$.

We believe that the updated RSLR dataset should address the referee's concern on RSLR.

Furthermore, we also updated our results and model based on the new RSLR dataset. In details, we found that the regional RSLR average reported from the 334 compiled sites was similar to what we have reported in the previous version (see Table 1). The result in Table 2 was not affected by the updated RSLR data. In Table 3, the correlation coefficients between C accumulation rate (CAR) and RSLR, and between sediment accretion rate (SAR) and RSLR were updated based on the new RSLR data. Both CAR and SAR were significantly positively associated with RSLR, similar to the results in the previous version. Fig. 1 and

Fig. 2 were not affected by the new RSLR data. Fig. 3 was updated based on the new RSLR dataset, and the number of submerging sites increased from 112 to 123 sites in Fig. 3a. In Fig. 3b, the relationship between SAR and RSLR was similar to the previous version.

New Fig 3 based on updated RSLR:

Old Fig 3:

In Fig. 7, the prediction of the conterminous U.S. coastal wetland annual C sequestration amount revised based on the new RSLR dataset, and the patterns are similar to the previous prediction. See below.

Old version Fig 7:

New version Fig 7:

In summary, we have updated the RSLR data based on the suggestions from Referee #1 and revised our result section based on the new dataset. We found that the application of the new RSLR dataset did not greatly change the relationships between CAR and RSLR and between SAR and RSLR. Moreover, the prediction of future C sequestration amount through 2100 under different climate change scenarios was not greatly affected by the new RSLR data.

Reference:

Holgate, S. J., A. Matthews, P. L. Woodworth, L. J. Rickards, M. E. Tamisiea, E. Bradshaw, P. R. Foden, K. M. Gordon, S. Jevrejeva and J. Pugh (2013). "New Data Systems and Products at the Permanent Service for Mean Sea Level." Journal of Coastal Research **29**(3): 493-504.

Reviewer #2 (Remarks to the Author):

I've read the revised manuscript and the authors response to the reviewers comments and am satisfied with the corrections the authors have made. Overall, the paper is much improved and I recommend it for publication.

Responses to Referee #2: Thanks. We hope that the updated version could meet the high standard of Nature Communications.

Reviewers' comments:

Reviewer #1 (Remarks to the Author):

This is my third review of the paper. I am pleased to see the authors have decided to standardized the data to the last 60 years. It is pleasing to see the effort put in by the authors to recalculate the relative sea level data. But a reference is need to support this timescale. They can use Engelhart et al., 2009. Geology or Douglas, 1992.

I am interested in the authors using the linear relationship between 30 and 60 years of records. I am surprised this is so consistent. I would like th authors to justify this relationship as it is very important for their calculated rates that goes to the heart of the publication

However, I think the authors should reflect on the new analysis. I am concerned by the authors' statement that the conclusions of the paper are "not greatly affected by the new RSLR data". First of all, what does "not greatly" actually mean?

Indeed if I look at Fig 7 (old vs new - despite having a different scaling for y axis) there does seem to be significant difference in the distribution of accretion vs sedimentation and RSLR.

But on a more serious point I would have thought there would be significant changes at certain locations. I ask this from examining the Table S3. For example why does a difference over of 6 mm/yr (#495, SKAGWAY) not influence the conclusions of the paper? There are several greater that 3 mm/yr. The large differences is why I suggested this standardization. I would like the authors to address why such large differences in the RSLR data do not influence the conclusions.

Response to comments

Editor's comments:

The referees' reports seem to be quite clear. Naturally, we will need you to address all of the points raised. Specifically, for publication in Nature Communications to be appropriate, we will need you to provide more compelling support for the statement that results are not greatly affected by the new RSLR data (Reviewer #1). You will also need to justify the linear relationship between 30 and 60 years of records (Reviewer #1).

Response: Thank you for permitting us to address reviewer 1's concerns. We have carefully considered all comments and suggestions from the anonymous reviewer 1, which have been incorporated into the revised version of our manuscript. As a consequence, we feel our manuscript has improved substantially. We greatly appreciate the opportunity for us to provide detailed response and an improved version of our manuscript which we believe it addresses all of the reviewer's comments.

Below we respond to each comment individually.

As suggested, we double checked our RSLR data, and there were 92 tidal gauge sites in the U.S., with 20 sites located at Alaska and Hawaii. Since our study was only conducted along the tidal wetlands in the conterminous U.S. (CONUS), the Alaska and Hawaii RSLR data were thus not useful in this study, and have no effect on our conclusions. For instance, reviewer 1 pointed out that some sites have over 3 mm difference in RSLR between the old and updated versions, of which were all located in Alaska. Please see attached Table R1 below. This explains why the new RSLR data are consistent to previous version as the sites with the large RSLR differences are from Alaska.

Moreover, in this study, we concluded that C accumulation rate (CAR) of the conterminous U.S. tidal wetlands shows a strong resilience to sea level rise, and our results show that they will continue to be a significant C sink through this century. This conclusion was supported by our results that the CAR was positively related with RSLR (Table 3). We have updated our RSLR data based on standardized RSLR as suggested, and this new RSLR dataset greatly improved the reliability of our interpretation. Indeed, the positive relation between CAR and RSLR remains consistent. Similar patterns have also been observed in recent studies at Australia (Rogers *et al.* 2019). We believe that the consistent CAR vs RSLR relationship reflects the strong resilience of tidal wetland to sea level rise along the U.S. (CONUS) coastal wetlands, which is a highly important finding.

We re-ran the linear relationship model between 30 and 60 yrs records for 43 CONUS sites (with over 60 yrs records), and provide a new Fig R1 based on the CONUS sites, and attached the raw data and R code for the analysis. We believe that this new figure and raw data will justify the linear relationship between 30 and 60 yrs RSLR. We also

provide Fig R2 to prove that our previous RSLR dataset was well correlated with the new standardized RSLR data.

Please find our detailed response to each of the individual questions and comments from Reviewer 1 below.

Reviewer #1 (Remarks to the Author):

Comments 1: This is my third review of the paper. I am pleased to see the authors have decided to standardized the data to the last 60 years. It is pleasing to see the effort put in by the authors to recalculate the relative sea level data. But a reference is need to support this timescale. They can use Engelhart et al., 2009. Geology or Douglas, 1992.

Reponses: We appreciate the Reviewer's suggestions. We have cited the Douglas 1992 and 1997 in the text to support the at least 60 yrs RSLR timescale dataset. Please see line 355.

Comments 2: I am interested in the authors using the linear relationship between 30 and 60 years of records. I am surprised this is so consistent. I would like the authors to justify this relationship as it is very important for their calculated rates that goes to the heart of the publication.

Reponses: In this version, we double checked all our data and calculations. We re-analyzed the relationship for the conterminous US (CONUS) sites (n=43), and we can see that the linear relationship between 30 and 60 yrs of records in our new figure (See Fig R1), which confirms this correlation. This relationship was developed based on the sites which have over 60 yrs records, for a total 43 sites in CONUS. The records along these relatively long-term sites is well supported by this linear relationship. We then used this relationship to standardize the RSLR of the other 29 sites which have less than 60 yrs but over 30 yr records in the CONUS.

To make it clearer, we also compared our full-time range RSLR and new standardized 60 yrs RSLR for the CONUS sites with time range between 60 yrs and 30 yrs (Fig R2), we can see that the full-time range RSLR correlate well with the standardized RSLR. This was mainly because most of these 29 sites have 40-59 yrs records (Table S3, n=21), which should well fit with the modeled 60 years RSLR. This also supports that the linear relationship that we observed between 30 and 60 yrs RSLR in the 43 long-term sites (over 60 yrs) which can be used for these short-term sites to be standardized to the 60 yrs RSLR.

To address these concerns and make our interpretations clear, we updated our Table S3 to include the detailed statistical P-values for each RSLR calculation in each site. We also attached the raw data for RSLR calculation which we extracted from the PSMSL database (see attached dataset). We also attached the R code to make these linear relationship analysis (see attached R code). From the raw data and R code, the

details of our 30 and 60 yrs RSLR rate calculations are clear. We feel that these raw data and R code justifies this relationship.

Fig R1. The linear relationship between recent 30 yrs RSLR and recent 60 yrs RSLR for the conterminous U.S. tidal gauge sites (CONUS) which have over 60 yrs records (n=43). The linear relationship ($y=0.85025x-0.57453$) was used to standardize the other 29 CONUS sites where the records ranged between 30 yrs and 60 yrs.

Fig R2. The relationship between old version RSLR and standardized RSLR for the sites with time records from 30 yrs to 60 yrs (n=29).

Comments 3: However, I think the authors should reflect on the new analysis. I am concerned by the authors' statement that the conclusions of the paper are "not greatly affected by the new RSLR data". First of all, what does "not greatly" actually mean? Indeed if I look at Fig 7 (old vs new - despite having a different scaling for y axis) there does seem to be significant difference in the distribution of accretion vs sedimentation and RSLR.

Reponses: We agree with the reviewer that the new RSLR data changed the distribution of RSLR vs soil accretion rate in Fig 3(a). In detail, the number of submerging sites increased from 112 sites in the old version to 130 sites in this updated version. However, in the both versions, most of the studied sites have higher accretion rate than their RSLR (n=213 for the updated dataset). To address your concerns at here, we attached more details of Fig 3(a) at below. Therefore, the updated RSLR dataset did not change this pattern.

Fig R3. More details of Fig 3a with 0.5 band width of X-axis.

Moreover, the positive correlations between soil accretion rate and RSLR (Fig 3b in the maintext) were consistent even when we updated the RSLR dataset. Although the new dataset did not change the patterns we have observed, the standardized RSLR data indeed improved our modeling and prediction. We appreciate the reviewer’s suggestion that has made us improve the quality of our manuscript.

Comments 4: But on a more serious point I would have thought there would be significant changes at certain locations. I ask this from examining the Table S3. For example why does a difference over of 6 mm/yr (#495, SKAGWAY) not influence the conclusions of the paper? There are several greater than 3 mm/yr. The large differences is why I suggested this standardization. I would like the authors to address why such large differences in the RSLR data do not influence the conclusions.

Response: In this study, we reported the conterminous U.S tidal wetlands, which did not include the tidal wetlands in Alaska and Hawaii. However, the 92 PSMSL tidal gauge stations dataset have all the sites in U.S., including 20 sites in Alaska and Hawaii. In the mentioned example (#495, SKAGWAY), the SKAGWAY site (59.45N, -135.327W) is located at Alaska. As a result, the variation of RSLR at this site did not affect our tidal wetland results in the conterminous U.S. We thus updated the Table S3 with 72 CONUS sites in this revised version (Table S3 in the supporting information), and labelled detailed location information for each of the stations. To address the reviewer’s comment and make our analyses clear, we have attached the previous version of Table S3 (92 sites including both CONUS sites and AK and HI sites, see below), and added detailed location of these sites (Table R1). We can see that all the sites which have over 2 mm difference in modeled RSLR and full records RSLR are located at Alaska, which highlights that the large difference of RSLR in these Alaska sites do not affect our data analysis in the conterminous US tidal wetlands.

Table R1. Previous version Tables S3 with updated location information for each sites. AK and HI sites was labeled with red color.

ID	Site	Location	La	Lo	Full	60yrs	30 yrs	Modeled	Used	Difference
					RSLR	RSLR	RSLR	RSLR	RSLR	In RSLR
225	KETCHIKAN	AK	55	-132	-0.4	-0.9	-1.5		-0.9	-0.51
405	JUNEAU	AK	58	-134	-13.3	-13.9	-14.8		-13.9	-0.58
426	SITKA	AK	57	-135	-2.4	-2.8	-3.5		-2.8	-0.31
445	YAKUTAT	AK	60	-140	-8.2	-10.2	-14.9		-10.7	-2.57
487	ADAK SWEEPER COVE	AK	52	-177	-1.4	-2.6	-2.7		-2.6	-1.23
495	SKAGWAY	AK	59	-135	-17.8	-18.6	-21.4		-15.4	2.43
757	UNALASKA	AK	54	-167	-4.5	-4.4	-1.1		-4.4	0.11
266	SEWARD	AK	60	-149	-2.8		-4.2	-3.1	-3.1	-0.30
1067	ANCHORAGE	AK	61	-150	-1.5		-3.5	-2.7	-2.7	-1.14
1070	SELDOVIA	AK	59	-152	-9.9		-8.6	-6.2	-6.2	3.66
1350	NIKISKI, ALASKA	AK	61	-151	-10.9		-12.3	-8.9	-8.9	1.99
1353	VALDEZ	AK	61	-146	-6.0		-8.8	-6.4	-6.4	-0.42
567	KODIAK ISLAND, WOMENS BAY	AK	58	-153	-8.9		-8.4	-6.1	-6.1	2.82
1634	SAND POINT, POPOF IS., AK	AK	55	-161	2.2		2.6	1.7	1.7	-0.46
10	SAN FRANCISCO	CONUS	38	-122	1.5	1.8	2.3		1.8	0.31
12	NEW YORK (THE BATTERY)	CONUS	41	-74	2.9	3.2	4.5		3.2	0.34
112	FERNANDINA BEACH	CONUS	31	-81	2.1	2.6	2.5		2.6	0.47
127	SEATTLE	CONUS	48	-122	2.1	2.0	2.8		2.0	-0.10
135	PHILADELPHIA (PIER 9N)	CONUS	40	-75	3.0	3.9	5.2		3.9	0.94
148	BALTIMORE	CONUS	39	-77	3.2	3.3	4.6		3.3	0.12
158	SAN DIEGO (QUARANTINE STATION)	CONUS	33	-117	2.2	2.4	3.3		2.4	0.21
161	GALVESTON II, PIER 21, TX	CONUS	29	-95	6.5	6.8	6.8		6.8	0.36
180	ATLANTIC CITY	CONUS	39	-74	4.1	4.5	4.8		4.5	0.35
183	PORTLAND (MAINE)	CONUS	44	-70	1.9	1.6	3.3		1.6	-0.28
188	KEY WEST	CONUS	25	-82	2.4	2.9	4.0		2.9	0.49
224	LEWES	CONUS	39	-75	3.4	3.8	5.5		3.8	0.33

	(BREAKWATER HARBOR)									
235	BOSTON	CONUS	42	-71	2.8	2.8	4.9		2.8	-0.03
234	CHARLESTON I	CONUS	33	-80	3.3	3.4	4.5		3.4	0.16
245	LOS ANGELES	CONUS	34	-118	1.0	1.4	2.3		1.4	0.35
246	PENSACOLA	CONUS	30	-87	2.4	2.7	5.2		2.7	0.35
256	LA JOLLA (SCRIPPS PIER)	CONUS	33	-117	2.0	2.1	1.8		2.1	0.04
299	SEWELLS POINT, HAMPTON ROADS	CONUS	37	-76	4.6	5.1	6.7		5.1	0.44
311	ANNAPOLIS (NAVAL ACADEMY)	CONUS	39	-76	3.6	3.7	5.4		3.7	0.09
332	EASTPORT	CONUS	45	-67	2.2	1.6	3.7		1.6	-0.53
351	NEWPORT	CONUS	42	-71	2.8	2.9	4.4		2.9	0.18
360	WASHINGTON DC	CONUS	39	-77	3.2	3.5	4.9		3.5	0.28
366	SANDY HOOK	CONUS	40	-74	4.1	4.1	5.9		4.1	0.00
367	WOODS HOLE (OCEAN. INST.)	CONUS	42	-71	2.9	3.0	5.2		3.0	0.15
377	SANTA MONICA (MUNICIPAL PIER)	CONUS	34	-119	1.6	1.3	2.5		1.3	-0.21
378	CRESCENT CITY	CONUS	42	-124	-0.8	-0.8	-1.0		-0.8	-0.07
384	FRIDAY HARBOR (OCEAN. LABS.)	CONUS	49	-123	1.2	1.2	2.3		1.2	0.00
385	NEAH BAY	CONUS	48	-125	-1.7	-2.1	-1.5		-2.1	-0.36
395	FORT PULASKI	CONUS	32	-81	3.2	3.6	4.5		3.6	0.39
396	WILMINGTON	CONUS	34	-78	2.4	2.9	4.2		2.9	0.51
412	SOLOMON'S ISLAND (BIOL. LAB.)	CONUS	38	-76	3.8	4.1	5.8		4.1	0.30
428	CEDAR KEY II	CONUS	29	-83	2.1	2.6	4.5		2.6	0.55
429	NEW LONDON	CONUS	41	-72	2.7	2.9	4.7		2.9	0.27
430	PROVIDENCE (STATE PIER)	CONUS	42	-71	2.3	2.4	4.5		2.4	0.15
437	ALAMEDA (NAVAL AIR STATION)	CONUS	38	-122	0.8	0.8	1.4		0.8	0.03
497	PORT ISABEL	CONUS	26	-97	4.0	4.5	5.7		4.5	0.48
508	PORT SAN LUIS	CONUS	35	-121	1.0	0.8	1.8		0.8	-0.14

316	MAYPORT	CONUS	30	-81	2.4	2.5			2.5	0.11
520	ST. PETERSBURG	CONUS	28	-83	2.8	3.1	4.1		3.1	0.31
526	GRAND ISLE	CONUS	29	-90	9.1	9.2	8.3		9.2	0.08
519	MONTAUK	CONUS	41	-72	3.2	3.5	5.4		3.5	0.33
525	BAR HARBOR, FRENCHMAN BAY, ME	CONUS	44	-68	2.3	2.0	3.2		2.0	-0.21
538	ROCKPORT	CONUS	28	-97	5.6	6.5	9.3		6.5	0.86
362	WILLETS POINT	CONUS	41	-74	2.4	2.1			2.2	-0.24
636	KIPTOPEKE BEACH	CONUS	37	-76	3.6	3.7	5.0		3.7	0.10
1068	BRIDGEPORT	CONUS	41	-73	2.9		4.3	2.9	2.9	0.03
1111	NANTUCKET ISLAND	CONUS	41	-70	3.7		5.1	3.5	3.5	-0.22
1106	FORT MYERS	CONUS	27	-82	3.2		5.4	3.7	3.7	0.56
1107	NAPLES	CONUS	26	-82	2.8		4.8	3.3	3.3	0.52
1153	CAPE MAY	CONUS	39	-75	4.6		6.1	4.2	4.2	-0.42
597	GLOUCESTER POINT	CONUS	37	-77	3.6		6.2	4.3	4.3	0.68
1156	DAUPHIN ISLAND	CONUS	30	-88	3.7		5.1	3.5	3.5	-0.25
1196	SOUTH BEACH	CONUS	45	-124	1.7		2.8	1.8	1.8	0.10
1193	APALACHICOLA	CONUS	30	-85	2.6		4.4	3.0	3.0	0.38
1295	CAMBRIDGE II	CONUS	39	-76	4.0		5.2	3.6	3.6	-0.41
2324	LEWISSETTA, VIRGINIA	CONUS	38	-76	5.5		6.9	4.7	4.7	-0.75
1325	PORT TOWNSEND	CONUS	48	-123	1.9		2.2	1.5	1.5	-0.40
2295	BEAUFORT, NORTH CAROLINA	CONUS	35	-77	3.7		4.6	3.1	3.1	-0.63
1352	MONTEREY	CONUS	37	-122	1.5		2.0	1.3	1.3	-0.26
1696	LAKE WORTH PIER	CONUS	27	-80	3.7		4.4	3.0	3.0	-0.70
1394	POINT REYES	CONUS	38	-123	2.1		3.1	2.1	2.1	-0.04
2125	ARENA COVE, CALIFORNIA	CONUS	39	-124	0.8		1.2	0.7	0.7	-0.14
2330	PORT CHICAGO, CALIFORNIA	CONUS	38	-122	1.9		3.3	2.2	2.2	0.31
1444	SPRINGMAID PIER	CONUS	34	-79	2.6		2.4	1.6	1.6	-1.03
2215	BAY WAVELAND	CONUS	30	-89	4.8		5.2	3.5	3.5	-1.23

	YACHT CLUB II									
1641	PANAMA CITY, ST.ANDREWS BAY, FL	CONUS	30	-86	3.9		4.4	3.0	3.0	-0.86
1636	DUCK PIER OUTSIDE	CONUS	36	-76	5.0		5.3	3.7	3.7	-1.34
1637	BERGEN POINT, STATEN IS.	CONUS	41	-74	4.8		5.2	3.6	3.6	-1.22
1638	CLEARWATER BEACH	CONUS	28	-83	4.7		6.0	4.2	4.2	-0.55
1639	N. SPIT, HUMBOLDT BAY	CONUS	41	-124	5.5		5.6	3.8	3.8	-1.68
786	REEDY POINT	CONUS	40	-76	4.0		4.4	3.0	3.0	-1.00
1635	CHESAPEAKE BAY BR. TUN.	CONUS	37	-76	6.0		6.4	4.4	4.4	-1.60
155	HONOLULU	HI	21	-158	1.5	1.4	1.9		1.4	-0.12
300	HILO, HAWAII ISLAND	HI	20	-155	3.2	2.7	3.1		2.7	-0.48
521	KAHULUI HARBOR, MAUI ISLAND	HI	21	-156	2.1	1.9	2.1		1.9	-0.20
756	NAWILIWILI BAY, KAUAI ISLAND	HI	22	-159	1.7	1.6	2.4		1.6	-0.11
823	MOKUOLOE ISLAND	HI	21	-158	1.5	1.5	2.2		1.5	0.00

To address the reviewer's concern, we re-analyzed the RSLR data based on the 72 CONUS sites. We can see that there was very good linear relationship between recent 60 yrs RSLR and full-time records RSLR in the 43 CONUS long-term sites (Fig R4). We further calculated the standardized RSLR and we can see that the standardized RSLR well match our full-time recorded RSLR (Fig R5 and Table S3, n=72). In Table S3, we also calculated the difference between the two RSLR dataset, and the maximum difference is 1.14 mm. We believe that these results can explain why the updated RSLR dataset did not change our conclusions.

Fig R4. The linear relationship between long-term RSLR and recent 60 yrs RSLR for the CONUS tidal gauge sites whose records were over 60 yrs (n=43).

Fig R5. The comparison of previous RSLR and standardized recent 60 yrs RSLR for 72 CONUS tidal gauge sites.

References:

Rogers, K., Kelleway, J.J., Saintilan, N., Megonigal, J.P., Adams, J.B., Holmquist, J.R., Lu, M., Schile-Beers, L., Zawadzki, A., Mazumder, D. & Woodroffe, C.D. (2019) Wetland carbon storage controlled by millennial-scale variation in relative sea-level rise. *Nature*, 567, 91-95.

REVIEWERS' COMMENTS:

Reviewer #1 (Remarks to the Author):

This is my forth review of the paper!

I have never done this before. Again, I am pleased to see the authors have further standardized the data.

I would suggest the authors should put a caveat regarding using linear relations to extend the 30 year records. I am still worried about the lack of impact of using a 30 year record. Surely there must be some stations where this relationship is in error

Reviewer #1 (Remarks to the Author):

This is my forth review of the paper!

I have never done this before. Again, I am pleased to see the authors have further standardized the data.

I would suggest the authors should put a caveat regarding using linear relations to extend the 30 year records. I am still worried about the lack of impact of using a 30 year record. Surely there must be some stations where this relationship is in error.

Response: We greatly appreciate the reviewer's effort to help us improve the manuscript. We added a sentence in the methods section to address this concerns. Please see line 441: ""We would like to point out that extending the 30 yr records using this linear relationship may not fit all stations, as some sites may have differing patterns.